

# Non-linear retreat of Jakobshavn Isbræ since the Little Ice Age controlled by geometry

Nadine Steiger[1], Kerim H. Nisancioglu[2], Henning Åkesson[2], Basile de Fleurian[2], and Faezeh M. Nick[3]

[1]Geophysical Institute, University of Bergen and Bjerknes Centre for Climate Research, Bergen, Norway
[2]Department of Earth Science, University of Bergen and Bjerknes Centre for Climate Research, Bergen, Norway
[3]University Centre in Svalbard, Lonyearbyen, Norway

*Correspondence to:* N. Steiger (nadine.steiger@student.uib.no)

**Abstract.** Rapid acceleration and retreat of Greenland's marine-terminating glaciers during the last two decades have initiated questions on the trigger and processes governing observed changes. Destabilization of these glaciers coincides with atmosphere and ocean warming, which broadly has been used to explain the rapid changes. To assess the relative role of external forcing versus fjord geometry, we investigate the retreat of Jakobshavn Isbræ in West Greenland, where margin positions exist since the Little Ice Age maximum in 1850. We use a one-dimensional ice flow model and isolate geometric effects on the retreat using a linear increase in external forcing.

We find that the observed retreat of 43 km from 1850 until 2014 can only be simulated when multiple forcing parameters—such as hydrofracturing, submarine melt and frontal buttressing by sea ice—are changed simultaneously. Surface mass balance, in contrast, has a negligible effect. While changing external forcing initiates retreat, fjord geometry controls the retreat pattern. Basal and lateral topography govern shifts from temporary stabilization to rapid retreat, resulting in a highly non-linear glacier response. For example, we simulate a disintegration of a 15 km long floating tongue within one model year, which dislodges the grounding line onto the next pinning point. The retreat pattern loses complexity and becomes linear when we artificially straighten the glacier walls and bed, confirming the topographic controls.

For real complex fjord systems such as Jakobshavn Isbræ, geometric pinning points predetermine grounding line stabilization and may therefore be used as a proxy for moraine build-up. Also, we find that after decades of stability and with constant external forcing, grounding lines may retreat rapidly without any trigger. This means that past changes may precondition marine-terminating glaciers to reach tipping-points, and that retreat can occur without additional climate warming. Present-day changes and future projections can therefore not be viewed in isolation of historic retreat.

## 1   Introduction

Marine-terminating glaciers export ice from the interior of the Greenland Ice Sheet (GrIS) through deep troughs terminating in fjords. Ice discharge occurs through ice dynamics and surface processes, each accounting for about half of the current GrIS mass loss (Khan et al., 2015). Since the beginning of the 21st century, net mass loss has doubled to a rate of $359\,\mathrm{Gt\,yr^{-1}}$ (2009–2012; Khan et al., 2014), due to increased surface runoff and acceleration of marine-terminating glaciers. While fast-



flow marine-terminating glaciers situated in southeastern Greenland have decelerated again, those in the northwestern part have not slowed-down yet (Moon et al., 2012).

Glacier dynamics is impacted by several processes linked to air and ocean temperatures, of which most are poorly understood as well as spatially and temporally heavily undersampled. A warmer atmosphere enhances runoff and causes crevasses to penetrate deeper through hydrofracturing, promoting iceberg calving (Benn et al., 2007; van der Veen, 2007; Cook et al., 2012, 2014; Pollard et al., 2015). A warmer ocean strengthens submarine melt below ice shelves and floating tongues (Holland et al., 2008a, b) that can reach melt rates up to $3.9\,\mathrm{m\,d^{-1}}$ during summer in western Greenland (Rignot et al., 2010). Turbulent melt water plumes thereby lift warm ocean water along submarine glacier faces (Jenkins, 2011), increasing submarine melt and calving rates (Luckman et al., 2015). This can potentially destabilize the glacier through longitudinal dynamic coupling and upstream propagation of thinning (Nick et al., 2009; Felikson et al., 2017). Increased air and fjord temperatures can additionally weaken sea ice and ice mélange in fjords, affecting calving through altering the stress balance at the glacier front (Amundson et al., 2010; Robel, 2017).

Observed acceleration and retreat around the GrIS is broadly consistent with large-scale atmospheric and oceanic warming (Carr et al., 2013; Straneo et al., 2013). Notwithstanding widespread acceleration, individual glaciers correlate poorly with regional trends (Moon et al., 2012) and only four glaciers have accounted for 50 % of dynamic mass loss since 2000, where Jakobshavn Isbræ (JI) in West Greenland has been the largest contributor (Enderlin et al., 2014). These heterogeneous patterns are poorly understood, inhibiting robust projections of sea level rise from marine ice sheet loss. Attribution of observed changes also remain challenging because the relatively short period of observational records constrains the understanding of the response of marine-terminating glaciers to external forcing. Here we therefore expand the range of climatic conditions to the period from the Little Ice Age (LIA) maximum in the mid-19th century to present-day.

Compared to previous studies, our focus on a longer time period provides context to recent observed changes on JI where our study is set. Since the LIA maximum, JI has retreated several tens of kilometres and accelerated significantly over the last two decades to become the fastest glacier in Greenland with a maximum velocity of $18\,\mathrm{km\,yr^{-1}}$ (measured in summer 2012; Joughin et al., 2014). The glacier's speed tripled within 20 years ($5.7\,\mathrm{km\,yr^{-1}}$ in 1992; Joughin et al., 2014), accompanied by thinning rates of $15\,\mathrm{m\,yr^{-1}}$ between 2003 and 2012 (Thomas et al., 2003; Krabill et al., 2004; Nielsen et al., 2013) and a doubling of ice discharge to a rate of about $50\,\mathrm{km^3\,yr^{-1}}$ (Joughin et al., 2004). JI alone contributed to 4 % of the global sea level rise in the 20th century (IPCC, 2001) and is the glacier in Greenland with the largest sea level contribution (Enderlin et al., 2014). It is also one of the most vulnerable glaciers in Greenland, with recent thinning potentially propagating as far inland as one third of the distance across the entire ice sheet (Felikson et al., 2017).

The destabilization of JI and other marine-terminating glaciers has been explained by regional warming (Holland et al., 2008a; Lloyd et al., 2011; Vieli and Nick, 2011; Carr et al., 2013; Straneo and Heimbach, 2013; Pollard et al., 2015). It is however well-known that grounding line stability is highly dependent on trough geometry, with landward sloping glacier beds potentially causing unstable, irreversible retreat (Schoof, 2007). The impact of glacier width is less studied, but lateral buttressing (Gudmundsson et al., 2012; Schoof et al., 2017) and topographic bottlenecks (Jamieson et al., 2012; Enderlin et al., 2013b; Jamieson et al., 2014) are suggested to stabilize grounding lines on reverse bedrock slopes. The dependence of ice discharge





and marine ice sheet stability on the subglacial and lateral topography implies different responses of individual glaciers, even if exposed to the same climate (Warren, 1991; Moon et al., 2012). This needs to be considered when inferring information on the climate by looking at glacier retreat reconstructions. Enderlin et al. (2013a) also showed that non-unique parameter combinations can exist for the same front positions, implying that real-world observations are vital to reduce uncertainty in

transient model simulations. However, very limited knowledge exists (Lea et al., 2014; Jamieson et al., 2014) regarding the interplay between bedrock geometry, channel-width variations and external controls on a real glacier.

In this study, we use a simple numerical ice flow model (Vieli et al., 2001; Nick et al., 2010; Vieli and Nick, 2011; Nick et al., 2013) to assess the relative impact of geometry and climate forcing on the observed retreat of JI from the LIA maximum to present-day. Geometric controls are isolated by using a linear forcing and the simplified model is validated using observed

velocities and front positions. Historical images and satellite observations provide detailed information on the non-linear frontal retreat of 43.2 km from the LIA position to the 2015 front position (Bauer, 1968; Stove et al., 1983; Sohn et al., 1997; Weidick et al., 2004; Alley et al., 2005). The aim of the study is here not to reconstruct the exact retreat history, but rather to study the external, glaciological and geometric controls on the model glacier in response to a linear forcing.

In Sect. 2 of this paper, knowledge on JI's retreat history, the local climate, and observations on its dynamics are reviewed.

This information is used to validate and realistically set up the model experiments. The experiments are run with a numerical ice flow model which is described generally in Sect. 3. The specific model setup used for the simulations is presented in Sect. 4, such as the used geometry, spin-up and initialization as well as the design of the conducted forcing and geometry experiments. In Sect. 5, the simulation results are revealed, starting with the LIA initial glacier, followed by the response of the glacier to the linear change in external forcing. The glacier response is thereby split up in the effect of different forcing parameter

combinations and the effect of simplified fjord geometries. Section 6, stresses the importance of the geometry compared to climate and discusses the resulting implication for geomorphology. Also the limitations of the simulations are disclosed, followed by a proposition on JI's future as suggested by the model simulations. Finally, Sect. 7 concludes the paper.

## 2  Jakobshavn Isbræ

JI is the fastest and most active glacier on the GrIS (Legarsky and Gao, 2006) and has been studied extensively during the last

decades. We use this relatively well-observed glacier to analyse the controls of the geometry and external forcing on its rapid retreat since the LIA. Observations that are used for the initialization and validation of the model are provided in this section.

### 2.1  Glacier setting and retreat history

JI is situated in West Greenland inland of Disko Bugt and terminates 60 km east of the town Ilulissat (see Fig. 1). About 6.5 % of the GrIS is drained through this outlet (Echelmeyer et al., 1991), with an annual ice discharge of about 40–50 km³ (Joughin

et al., 2004; Cassotto et al., 2015), producing 10 % of all icebergs released from the GrIS (Weidick and Bennike, 2007). The lowermost 70 km of the glacier rest on a deeply incised trough partially exceeding a depth of 1 km and flows through a narrow ($< 5$ km wide) subglacial trough (Morlighem et al., 2014).



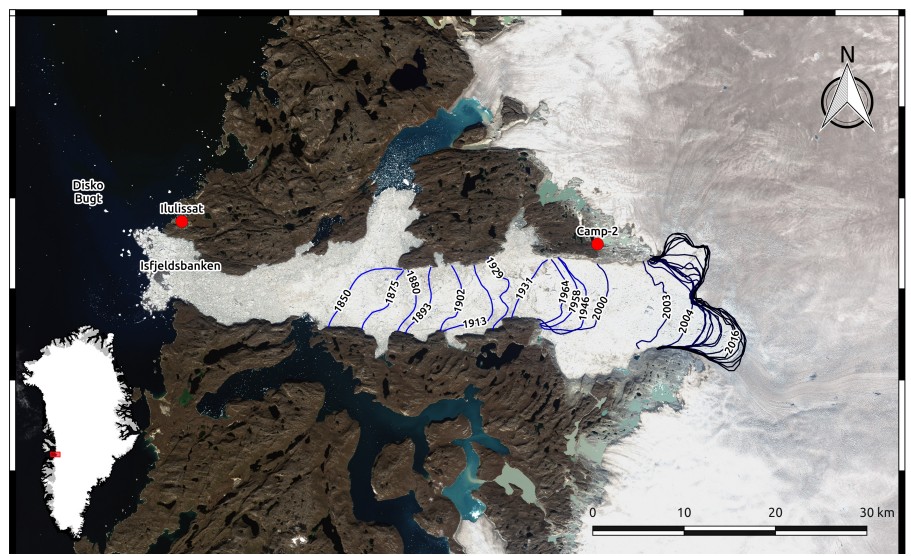

**Figure 1.** Glacier front positions of JI from Khan et al. (2015) (1850–1985) and CCI products derived from ERS, Sentinel-1 and LANDSAT data by ENVO (1990–2016). The background map is a LANDSAT-8 image from 16 August 2016 (courtesy of the U.S. Geological Survey). Locations names that occur in the text are marked. The inset shows the location of JI on Greenland.

Radiocarbon dating suggests that Disko Bugt deglaciated rapidly between 10.5–10.0 thousand years before present (kyr BP) from the outer coast (Ingölfsson et al., 1990; Long et al., 2003) until the ice sheet margin reached a sill (Isfjeldsbanken) at Ilulissat (see Fig. 1) at about 9.5 kyr BP. A period of little retreat resulted in the formation of large moraine systems (Weidick and Bennike, 2007). Thereafter, the front retreated to its 1960 position around 6 kyr BP with further 20 km retreat until ca.
4–5 kyr BP (Weidick et al., 1990). Subsequent cooling during the LIA (1500–1900) initiated an advance to the LIA maximum position around year 1850, about 43 km downstream of the present-day position (Weidick and Bennike, 2007). After 1850, the glacier retreated again by 30 km and was relatively stable between 1960 and 1992, only fluctuating seasonally by 2.5 km (Echelmeyer et al., 1991; Sohn et al., 1998). This stabilization was accompanied by a slow-down during 1985 to 1992 (Joughin et al., 2004). In 1991, the glacier even readvanced 3 km and thickened at lower elevations by more than $1\,\mathrm{m\,yr^{-1}}$ until 1997,
followed by a thinning of up to $10\,\mathrm{m\,yr^{-1}}$ after 1998 (Abdalati et al., 2001; Thomas et al., 2003). Due to the thinning, the 15 km long floating tongue weakened and disintegrated completely from 2001 until May 2003 (Thomas et al., 2003; Joughin et al., 2004; Luckman and Murray, 2005). This rapid retreat and loss of the floating tongue initiated a doubling of velocities (Joughin et al., 2004) and a rapid, still ongoing retreat. In summer 2012, JI reached its maximum terminus velocity of up to $18\,\mathrm{km\,yr^{-1}}$ (Joughin et al., 2014). Future simulations suggest a possible further retreat of about 40 km until 2200 (Nick et al.,
2013).



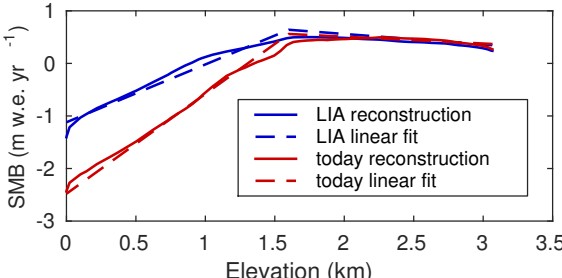

**Figure 2.** SMB profiles along JI's main flowline at LIA (1840–1850 average) and present-day (2002–2012 average) from reconstructions by Box (2013) and the linear fit used in the model.

## 2.2 Atmospheric and oceanic conditions

Monthly surface mass balance (SMB) of the GrIS since the LIA (1840–2012) is reconstructed by Box (2013) using a combination of meteorological station records, ice cores, regional climate model output and a positive-degree day model. The data shows that the SMB along JI is most negative at the margin and increases with height to a transition zone beyond which it

decreases again with height (Fig. 2). The low SMB of -1 to -3 m yr$^{-1}$ at the glacier front is due to high temperatures along the coast, whereas the decreasing SMB in the interior is due to diminishing precipitation. Since 1840, the SMB in the upper area has stayed at a similar level, whereas the lower 1.5 km have shown an interannual variability with a decreasing trend during the last two decades (Fig. 2, here only shown the SMB change between the LIA period and present-day).

In addition to its effect on mass balance, surface melting produces water that penetrates into crevasses and to the glacier

base. Surface melting on the GrIS has increased by 63 % from 1840 to 2010 (Box, 2013).

The retreat of JI during the last two decades is contemporary to increased ocean temperatures (Holland et al., 2008a; Lloyd et al., 2011) due to a warming of subpolar North Atlantic water flowing along Greenland's coast (Straneo and Heimbach, 2013). Lloyd et al. (2011) present subsurface ocean temperatures (at 300 m depth) at the western margin of Disko Bugt from benthic foraminiferal records. This reconstruction shows increased temperatures from 1920 to 1950 and since 1998; two warming

periods that are followed by a retreat of JI's calving front. Direct measurements of submarine melt rates are rare, but they are estimated to be up to two orders of magnitudes larger than surface melt rates and comparable to calving rates (Rignot et al., 2010). Annual submarine melt rates were about 228 m yr$^{-1}$ before the disintegration of the floating tongue (measured in 1985; Motyka et al., 2011). Jenkins (2011) estimates an increase in submarine melt rates by 30 % due to an observed ocean warming of about 1°C in 1997 (e.g. Holland et al., 2008a; Motyka et al., 2011; Hansen et al., 2012) and more than a doubling of melt rates

when additionally considering the steepening of the calving front as consequence of large calving events. Submarine melt may also be enhanced by an increase in subglacial discharge due to larger surface runoff (Jenkins, 2011; Xu et al., 2012; Sciascia et al., 2013; Xu et al., 2013), although this may only be a local effect in front of cavities and negligible when width-averaged (Cowton et al., 2015).



Sea ice in fjords adjacent to the calving front binds together ice bergs, forming ice mélange that prevents calving by buttressing and lengthens the floating tongue (Geirsdóttir et al., 2008; Amundson et al., 2010; Cassotto et al., 2015). This longer tongue decreases flow velocities along the glacier through longitudinal forces (Thomas, 1979; Nick et al., 2009). The seasonality of sea ice cover coincides with the seasonality of calving front migration; during winter, the front is held back from calving and grows longer, whereas the weakened ice mélange during summer causes a rapid retreat of the front. Especially JI terminates in a fjord that is densely filled with ice bergs unable to exit the fjord due to the shallow (200–300 m deep) Isfjeldsbanken (Weidick and Bennike, 2007). However, during summer when the sea ice melts, JI advances to up to 5 km (Cassotto et al., 2015). Increasing air and fjord temperatures shorten the time period of sea ice cover, so that calving is retained for a shorter period, decreasing the seasonality of calving rates (Amundson et al., 2008). The effect of reduced sea ice cover on annual calving fluxes is unknown.

## 2.3 Glacier dynamics

Surface velocities from satellite radar interferometry covering the GrIS are available from Rignot and Mouginot (2012). Additionally, continuous observations of the velocities of JI since 1992 exist at several positions close to the calving front, with continuous time series since 2005 (Joughin et al., 2004, 2012, 2014). Velocities display a strong seasonal cycle and reached their maximum of $18\,\mathrm{km\,yr^{-1}}$ in summer 2012 at the glacier front. The annual average in 2012 was about $12\,\mathrm{km\,yr^{-1}}$ (Joughin et al., 2014).

Basal motion can account for up to 90 % of surface velocities of temperate glaciers (Bamber et al., 2007) and is especially high for ice streams (Shapero et al., 2016). Nevertheless, the observed high velocities at JI were first believed to result from internal deformation, because basal resistance was thought to be very high (Iken et al., 1993; Lüthi et al., 2002; van der Veen et al., 2011). More recent ice flow simulations suggest low basal resistance (Joughin et al., 2012; Habermann et al., 2013) and data assimilation methods imply basal stresses at the bed of the deep trough of about 65 kPa at 50 km upstream of the calving front, equivalent to only 20 % of the driving stress (Shapero et al., 2016). These studies therefore imply that most resistance is given by the shear margins.

The high discharge rates observed at JI may have already been initiated around 8 kyr BP, when the glacier front retreated from Isfjeldsbanken near Ilulissat (Weidick and Bennike, 2007). Before the disintegration of the floating tongue in 2001, ice discharge was in the range of $21\text{–}27\,\mathrm{km^3\,yr^{-1}}$ (Echelmeyer et al., 1991, 1992; Sohn et al., 1998), and has almost doubled until the beginning of the 21st century to $35\text{–}50\,\mathrm{km^3\,yr^{-1}}$ (Joughin et al., 2004; Rignot and Kanagaratnam, 2006; Cassotto et al., 2015).





## 3 Modelling approach

### 3.1 Numerical ice flow model

We use a width- and depth integrated numerical ice flow model constructed for marine-terminating glaciers (Vieli et al., 2001; Vieli and Payne, 2005; Nick et al., 2009, 2010). Ice thickness changes with time are calculated from the along-flow ice flux and mass balance, using a width- and depth integrated continuity equation (Eq. 1).

$$\frac{\partial H}{\partial t} = -\frac{1}{W}\frac{\partial(HUW)}{\partial x} + \dot{B} \tag{1}$$

$H$ is thereby the thickness, $W$ the width, $U$ the velocity and $x$ the along-flow component. The mass balance $B$ includes SMB and submarine melt.

The ice flux is controlled by a balance of lateral and basal resistance, along-flow longitudinal stress gradient and driving stress (Eq. 2). Lateral resistance is parametrized using a width-integrated horizontal shear stress (van der Veen and Whillans, 1996) and we use a Weertman-type basal sliding law based on effective pressure (Fowler, 2010). The longitudinal stress gradient is dependent on the effective viscosity $\nu$, which is non-linearly dependent on the strain rate.

$$2\frac{\partial}{\partial x}\left(H\nu\frac{\partial U}{\partial x}\right) - A_s\left[\left(H - \frac{\rho_w}{\rho_i}D\right)U\right]^{1/m} - \frac{2H}{W}\left(\frac{5U}{E_{lat}AW}\right)^{1/n} = \rho_i g H \frac{\partial h}{\partial x} \tag{2}$$

$h$ is the surface elevation, $g$ the gravitational acceleration, $D$ is the depth of the glacier below sea level, $\rho_i$ and $\rho_w$ are the densities of ice and ocean water, respectively. $A$ is the rate factor and $n$ and $m$ are the exponents for Glen's flow law and sliding relations, respectively. The lateral enhancement factor $E_{lat}$ is used to tune the lateral resistance and the basal sliding parameter $A_s$ tunes the resistance from the bed.

The grounding line position is treated robustly and calculated with a flotation criterion (van der Veen, 1996). A moving uniform spatial grid is initiated with a grid size of $\Delta x = 300$ m which is adjusted to the new glacier length in each time step, keeping the number of gridpoints constant (Nick and Oerlemans, 2006). At the marine terminus, a dynamic crevasse-depth calving criterion is used and further explained in Sect. 3.2. The equations are solved by a Newton iteration method and computed on a staggered grid between the grid points.

### 3.2 Calving parametrization

The crevasse-depth criterion calculates calving where the surface crevasse depth ($cd_s$) and basal crevasse depth ($cd_b$) penetrates the whole glacier thickness (Nick et al., 2010). The depth of basal crevasses is calculated from tensile deviatoric stresses and the height above buoyancy (Eq. 3).

$$cd_b = \frac{\rho_i}{\rho_w - \rho_i}\left(\frac{R_{xx}}{\rho_i g} - (H - \frac{\rho_w}{\rho_i}D)\right) \tag{3}$$



**Table 1.** List of physical parameters and constants used in the model. Values for the rate factor ($A$) are taken from Cuffey and Paterson (2010). Values for the parameters changing with climate and used for the experiments are listed in Table 2.

| Parameter name | Symbol | Value | Unit |
|---|---|---|---|
| Gravitational acceleration | $g$ | 9.8 | $\mathrm{m\,yr^{-1}}$ |
| Ice density | $\rho_i$ | 900 | $\mathrm{kg\,m^{-3}}$ |
| Ocean water density | $\rho_w$ | 1028 | $\mathrm{kg\,m^{-3}}$ |
| Fresh water density | $\rho_{fw}$ | 1000 | $\mathrm{kg\,m^{-3}}$ |
| Sliding exponent | $m$ | 3 | |
| Glen's flow law exponent | $n$ | 3 | |
| Rate factor | $A$ | A(-20°C) – A(-5°C) | $\mathrm{yr^{-1}\,Pa^{-2}}$ |
| Basal resistance parameter | $A_s$ | 120 | $\mathrm{Pa\,m^{-2/m}\,s^{-1/m}}$ |
| Lateral enhancement | $E_{lat}$ | 10 | |
| Grid size | $dx$ | 250–300 | m |
| Time step | $dt$ | 0.005 | yr |

Opening of surface crevasses is caused by tensile deviatoric stresses and enhanced by melt water filling up crevasses due to the additional water pressure (Nye, 1957; Nick et al., 2013, Eq. 4):

$$cd_s = \frac{R_{xx}}{\rho_i g} + \frac{\rho_{fw}}{\rho_i} cwd, \tag{4}$$

where $cwd$ is the crevasse water depth, $\rho_{fw}$ and $\rho_i$ are the densities of freshwater and ice respectively. The tensile deviatoric stress $R_{xx}$ is the difference between tensile stresses that pull a fraction open and the ice overburden pressure. $R_{xx}$ is calculated from the longitudinal strain rate $\dot{\epsilon}_{xx}$ through Glen's flow law (Eq. 5).

$$R_{xx} = 2\left(\frac{\dot{\epsilon}_{xx}}{A}\right)^{1/n} \tag{5}$$

In the model, buttressing by sea ice is implemented as a factor ($f_{si}$), which can be reduced accounting for weakening of ice mélange by increasing the strain rate. The strain rate (Eq. 6) is responsible for the opening and downward-penetration of crevasses at the glacier terminus, consequently increasing calving rates.

$$\dot{\epsilon}_{xx} = \frac{\partial U}{\partial x} = \frac{1}{f_{si}} A \left[\frac{\rho_i g}{4}\left(H - \frac{\rho_w}{\rho_i}\frac{D^2}{H}\right)\right]^n \tag{6}$$

### 3.3 Atmospheric and ocean forcing

The observed along-flow SMB at JI (Fig. 2) can be approximated by a linear function of the surface elevation divided into two parts: the steep lower part below the transition height $h_0$ ($h(x) \leq h_0$) where the SMB increases with elevation and the upper





area of low precipitation ($h(x) > h_0$) where the SMB decreases with elevation (Eq. 7).

$$a(x) = \left(a_0 - \frac{da}{dx} \cdot h_0\right) + \frac{da}{dx} \cdot h(x); \text{ with } \frac{da}{dx} = \begin{cases} S_{a1} & \text{at } h(x) \leq h_0 \\ S_{a2} & \text{at } h(x) > h_0 \end{cases} \tag{7}$$

In Table 2, the values for the vertical gradients $S_{a1}$ and $S_{a2}$ as well as the SMB $a_0$ at the height $h_0$ are given.

Submarine melt is implemented in the model as a vertical melt rate applied where the glacier is floating. The submarine melt

rate may vary dependent on glacier discharge and the slope of the submarine face. However, the relationships are still poorly known and width-averaged submarine melt rates are even found to be relatively insensitive to glacier discharge (Cowton et al., 2015). In the model, a distance-dependent submarine melt rate profile can be implemented. We find that the response of the glacier is similar to using a constant value for submarine melt along the whole floating tongue (not shown here).

Additionally to the SMB, the glacier is fed by tributary ice flow mainly adding mass in the lowermost 80 km (see Fig. 4).

The lateral influx $Q_L$ is initially calculated at each grid point as the sum of both lateral fluxes (velocity $v_{L,0}$ times depth $H_{L,0}$), divided by the width of the main trough $W_{JI}$ (Eq. 8). This influx locally accounts to about 100 times the SMB, with a maximum of $120\,\text{m yr}^{-1}$. The influx changes with time, as the velocity and thickness changes. We assume that these changes are comparable to the changes of the main ice flow and scale it with the new velocity and thickness ($v_{JI,t}$ and $H_{JI,t}$) compared to the initial velocity and thickness ($v_{JI,0}$ and $H_{JI,0}$).

$$Q_{L,0}(x) = \frac{v_{L,0}(x) \cdot H_{L,0}(x)}{W_{JI}(x)}$$

$$Q_{L,t}(x) = Q_{L,0}(x) \cdot \frac{v_{JI,t}(x) \cdot H_{JI,t}(x)}{v_{JI,0}(x) \cdot H_{JI,0}(x)} \tag{8}$$

## 4   Model setup

The glacier model is initialized with observed bed topography data as well as the glacier extent during the LIA maximum (1850) and an observed trimline height (the surface elevation during the LIA) at Camp-2 (see Fig. 1). From this steady-state initial glacier, an increase in external forcing is applied to force the observed glacier retreat of 43.2 km between until 2015. The

initial parameters and their perturbations are constrained by the observations described in Sect. 2.

### 4.1   Initialization

Bathymetry data and subglacial bed topography data for JI are sparse and difficult to attain due to the abundance of ice mélange in the fjord and a sediment rich bed beneath the glacier. We use a one-dimensional along-flow bed topography profile in the deep trough and fjord as it is presented in Boghosian et al. (2015). The fjord bathymetry is thereby obtained from

Operation IceBridge gravity data, whereas the profile from high-sensitivity radar data by Gogineni et al. (2014) are favoured for the subglacial trough and used here. For the bed upstream of the deep trough (77 km from the 2015 front position), 150 m resolution data by Morlighem et al. (2014) are averaged over the glacier width. The glacier width is defined as the trough width

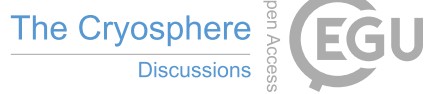



at the present-day sea level from topography data (Morlighem et al., 2014) and satellite images in the ice-free fjord (Fig. 1). JI's catchment widens gradually over the upper 445 km up to the ice divide, and the width is defined accordingly following Nick et al. (2013).

Basal sliding in the model decreases the surface slope and hence the ice thickness at the ice divide. The basal sliding parameter $A_s$ is therefore adjusted for the LIA to achieve an observed present-day thickness of 3065 m at the ice divide (Howat et al., 2014); the present-day thickness can be used because the ice sheet is assumed to be in steady-state above 2000 m of elevation (Krabill, 2000). As the impact of increased melt on basal sliding on interannual time scales is still unknown (Sole et al., 2011), the basal sliding parameter is kept constant in time and space in our model simulations.

The glacier surface is also determined by the lateral resistance. A lateral enhancement factor of $E_{lat} = 10$ is applied along the glacier to account for lateral heating and to achieve a present-day surface corresponding to observations (Howat et al., 2014).

Temperature profiles from boreholes at about 50 km upstream of the calving front show ice temperatures between -5°C and -22°C with the warmest temperatures at the surface and the bottom (Lüthi et al., 2002). Here, the ice temperature, used to determine the rate factor $A$, is depth-averaged and decreases from -5°C at the ice front to -20°C at the ice divide to account for colder air temperatures further inland. In the modelled glacier, the temperature in the narrow part at the glacier front largely determines the glacier surface, so that the temperature of -5°C (also used by Nick et al., 2009; Vieli and Nick, 2011; Nick et al., 2013, for JI) is dominant. Little change in ice temperature over the time scale can be expected (Seroussi et al., 2013) and we therefore keep the rate factor temporally constant.

The depth of water filling crevasses has not been measured yet, but the chosen value of 160 m for the steady-state achieves the observed glacier length and a calving rate of $34 \, \mathrm{km^3 \, yr^{-1}}$ in 1985, which is in the same order of magnitude as the observed calving rate of $26.5 \, \mathrm{km^3 \, yr^{-1}}$ in 1985 (Joughin et al., 2004).

Sea ice buttressing is initially set to $f_{si} = 1$ in the steady-state but reduced in some of the experiments to account for a weakening of ice mélange.

## 4.2 Forcing experiments

The simulated retreat of 43.2 km between 1850 and 2015 is reproduced by a linear increase in submarine melt rate, crevasse water depth and a reduction in sea ice buttressing. The SMB is well known (Box, 2013), and all runs are therefore forced with the same gradual change in the SMB gradients from the LIA (Fig. 2). For the other forcings, however, nine different parameter combinations are applied to cover a range of reasonable values that are reached at present-day. Table 2 shows the parameter values that are used for the steady-state glacier during LIA and those that are reached in 2015 after the gradual increase. Figure 3 illustrates the nine experiments and the corresponding present-day values for each of the parameters.

The sea ice buttressing can be assumed to be linearly dependent on the ocean and air temperatures (Nick et al., 2013), both of which have increased since the LIA. It may therefore have decreased by a factor up to three, which is used in Nick et al. (2013). However, a temperature increase may only have an impact on seasonal frontal migration, leaving the annual calving fluxes unaffected. We therefore conduct experiments with a sea ice buttressing of 100 %, 50 % and 33 % for present-day.



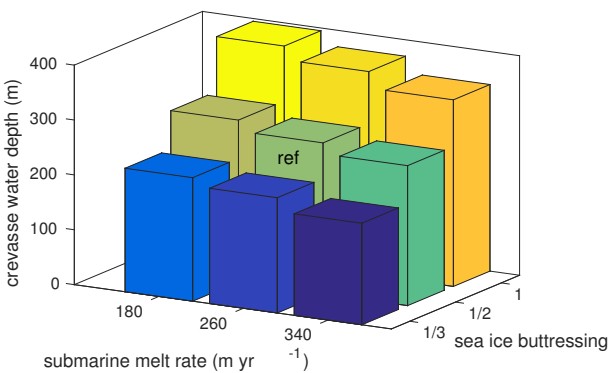

**Figure 3.** Sample of parameter combinations of submarine melt rate, crevasse water depth and sea ice buttressing that force the observed retreat of 43.2 km between the LIA and 2015. Shown values are those reached at present-day after a gradual linear increase. Colours indicate the different model runs. The reference best-guess run is marked here ("ref") and analysed in more detail in Sect. 5.2.

**Table 2.** Forcing parameters that are used in the model initially (LIA) and reached in 2015 after the linear perturbations.

| Parameter | Symbol | LIA value | Perturbation range for 2015 | Unit |
|---|---|---|---|---|
| Submarine melt rate | $smr$ | 175 | 175–340 | $\mathrm{m\,yr^{-1}}$ |
| Crevasse water depth | $cwd$ | 160 | 160–360 | m |
| Lower SMB gradient | $S_{a1}$ | 0.0011 | 0.0019 | $\mathrm{yr^{-1}}$ |
| Upper SMB gradient | $S_{a2}$ | -0.0002 | 0.00013 | $\mathrm{yr^{-1}}$ |
| Max SMB | $a_0$ | 0.64 | 0.56 | $\mathrm{m\,w.e.\,yr^{-1}}$ |
| Sea ice buttressing | $f_{si}$ | 1 | 1/3–1 | |

Submarine melt is influenced by ocean temperatures, which have increased by approximately 50 % since 1980 (Lloyd et al., 2011). Additionally, it is affected by subglacial discharge and turbulent mixing, which may have increased prior to 1980. Since submarine melt rate is poorly constrained as we go further back in time, we increase present-day melt rates to up to twice the LIA value.

5     The increase in crevasse water depth is unknown, but may be comparable to the increase in runoff, which has increased by 63 % since the LIA (Box, 2013). To account for a large range, we increase crevasse water depth within a range of 20 % to 100 % relative to the LIA. It is thereby tuned depending on the combination of sea ice buttressing and submarine melt rate to reach the observed retreat (Fig. 3). Figure 3 shows that the parameters are dependent on each other, meaning e.g. that a high submarine melt rate is needed in case of reduced sea ice buttressing and a small crevasse water depth or that the crevasse water
10 depth has to be large when sea ice buttressing is not reduced and the submarine melt rate small.

Despite the well-known retreat history of JI since the LIA (Fig. 1), we only constrain the forcing perturbations for the experiments by the observed LIA and present-day front positions, to isolate the geometric impact on the retreat. Nevertheless, we present the time evolution of the simulated front positions together with observations. To obtain one-dimensional front positions, we first calculate a centreline as a smoothed line following the mean latitudinal position of each observed glacier front (Fig. 1). The front positions are then chosen where the glacier fronts intersect with the centreline and plotted with the maximal spread of each front in longitudinal direction.

### 4.3 Geometric experiments

In addition to the effect of forcing, we also test the impact of fjord geometry on glacier retreat. We design experiments with a smoothed width and bed in the deep and narrow trough. Four different geometry combinations are constructed and shown in Fig. 4.

  **a** Original geometry: Observed width and bed of the trough as described in Sect. 4.1

  **b** Straight width: The width until 80 km inland of today's front is set to a constant value of 5.4 km. Only at the LIA front position, a wide section is kept in order to reach a steady-state with the same parameters. The bed is kept as in **a**.

  **c** Straight bed: The bed of the deep trough to 120 km inland of today's front is smoothed to get an almost straight bed, linearly rising inland. The width is kept as in **a**.

  **d** Straight width and bed: As a combination of the straightened bed and straightened width, both are straightened here.

The runs with simplified geometry start from steady-state at the LIA front position with the same parameters and forcing as for the original geometry (Table 2). Due to the changed topographies, the glacier surfaces and velocities differ from the original geometry and the LIA front position is slightly changed.

## 5 Results

In this section, we present the steady-state glacier at the LIA maximum extent and the simulated glacier retreat as response to the most reasonable forcing perturbation. The response is then compared to different forcing parameter combinations and more simplified geometries.

### 5.1 Jakobshavn Isbræ at the LIA maximum

The initial glacier is run to steady-state with known conditions for the LIA maximum and shown in Fig. 4a. It has an uneven surface which mirrors the trough geometry (Gudmundsson, 2003). The only constraint for the glacier thickness is given by the assumption of a balanced surface above 2000 m elevation (Krabill, 2000) and measurements of the LIA trimline at Camp-2 with a height of 300 m (Csatho et al., 2008, see Fig. 5). The position of Camp-2 is located on a modelled surface bump,





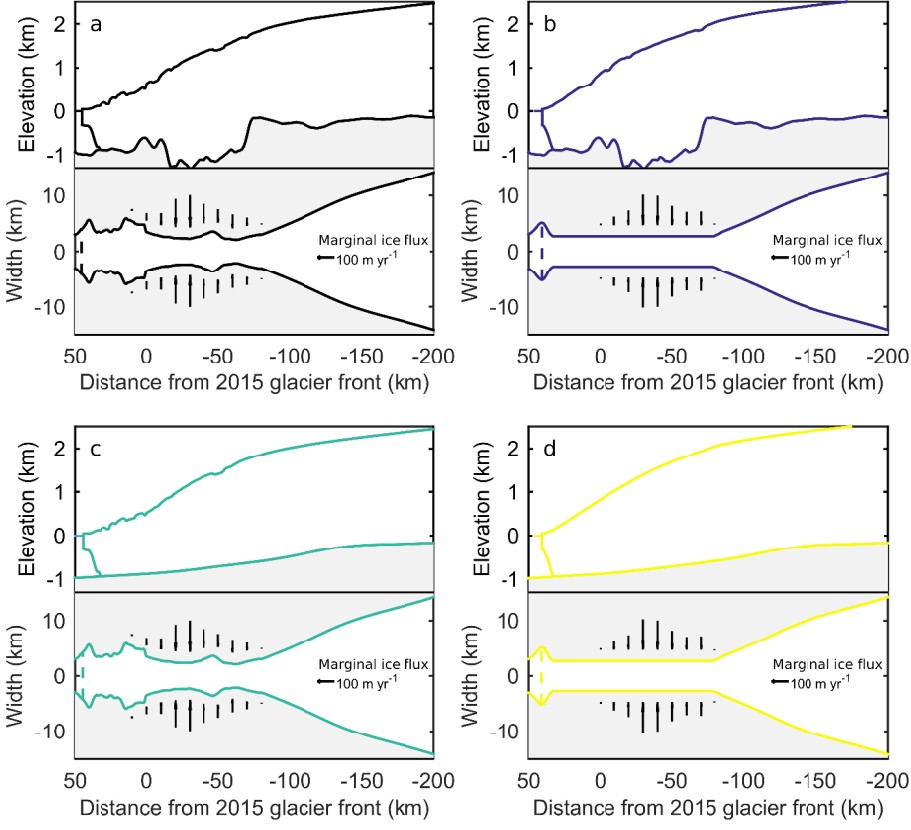

**Figure 4.** Different model geometries used to investigate the impact of topography on ice dynamics. (a) Original geometry, (b) straight width, (c) straight bed and (d) straight width and bed. Arrows indicate the tributary ice flux, with their length representative for the influx volume.

overestimating the thickness by $100\,\mathrm{m}$ compared to the observed trimline height from the LIA (Csatho et al., 2008); however, smoothing of the surface reduces this overestimation. The LIA glacier terminates with a $9\,\mathrm{km}$ long floating tongue, where it has a velocity of $5\,\mathrm{km\,yr^{-1}}$ and a volume flux of $35\,\mathrm{km^3\,yr^{-1}}$. The modelled width-averaged basal shear stress is about $128\,\mathrm{kPa}$ at $40\,\mathrm{km}$ inland of the present-day front position and the driving stress is $290\,\mathrm{kPa}$ at that location, when applying a $3\,\mathrm{km}$ moving
5  average to smooth the surface pumps.

## 5.2 Non-linear glacier response to linear forcing

Figure 5 shows the modelled glacier retreat from the LIA position forced by a linear increase in SMB, submarine melt rate, crevasse water depth and reduction in sea ice buttressing. Here, the model run with the best estimate of forcing parameters is presented (reference run in Fig. 3). The modelled front position retreats non-linearly in response to the linear external forcing.
10  It retreats $21\,\mathrm{km}$ during the first 163 years, after which a $16\,\mathrm{km}$ long floating tongue forms. During the break-off of the tongue in 2013 to 2014, the front retreats further $23\,\mathrm{km}$. Throughout the retreat, the glacier terminus changes between a floating tongue





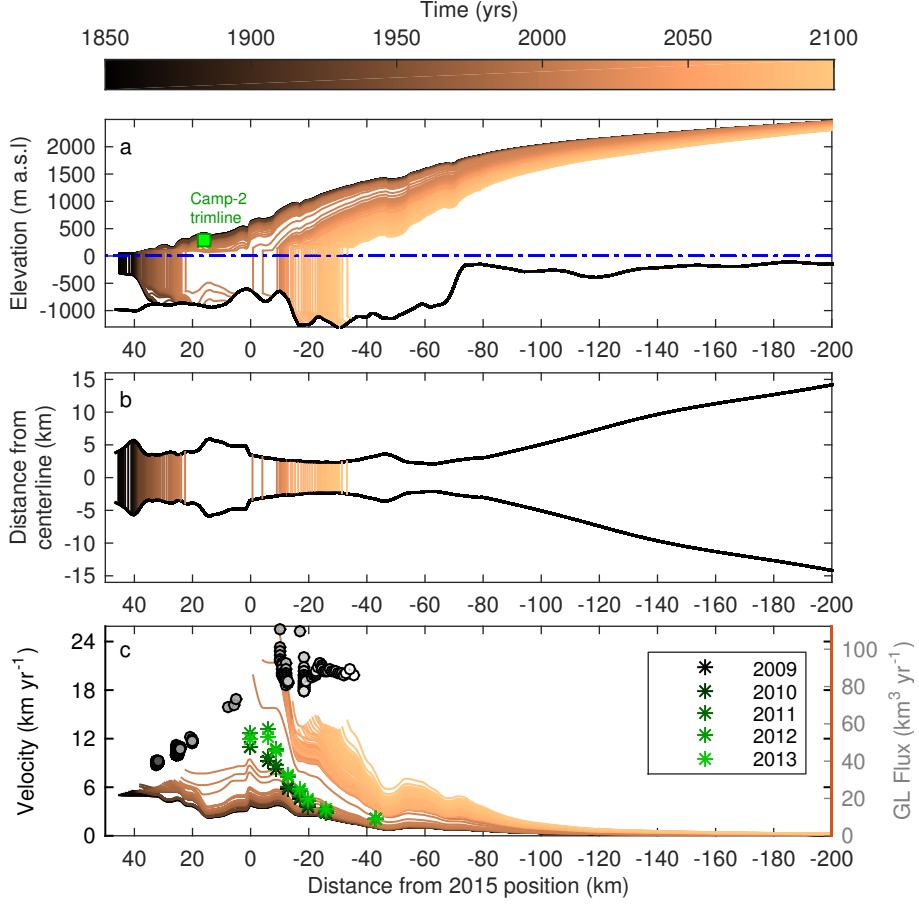

**Figure 5.** Modelled retreat of JI in response to a gradual change to the reference parameters (see Fig. 3 "ref"). Yearly profiles are shown for (a) the along-flow glacier profile and the height of the Camp-2 LIA trimline by Csatho et al. (2008) in green, (b) the front positions in a top-view and (c) the along-glacier velocities including the grounding line (GL) flux and observed yearly velocities at 7 different points upstream from the glacier front from 2009 to 2013 (Joughin et al., 2014). Shown are 300 model years.

and a grounded front. The front velocities only increase by $3 \, \mathrm{km} \, \mathrm{yr}^{-1}$ during the first 163 years and more than double from $8 \, \mathrm{km} \, \mathrm{yr}^{-1}$ to $19 \, \mathrm{km} \, \mathrm{yr}^{-1}$ when the floating tongue breaks off. This acceleration is overestimated, as the simulated tongue breaks off faster than observed. However, velocity observations by Joughin et al. (2014), shown in Fig. 5c, are in-between the simulated velocities before and after the break-off. The model simulations show that the acceleration continues until the retreat of the front slows down. The grounding line flux, calculated as the grounding line velocity times the grounding line gate area, increases from $35 \, \mathrm{km}^3 \, \mathrm{yr}^{-1}$ to $65 \, \mathrm{km}^3 \, \mathrm{yr}^{-1}$ from the LIA until 2015. Beyond 2015 it goes up to $100 \, \mathrm{km}^3 \, \mathrm{yr}^{-1}$ and finally stabilizes at $77 \, \mathrm{km}^3 \, \mathrm{yr}^{-1}$.

Many different parameter combinations presented in Fig. 3 lead to the observed total retreat since the LIA. Figure 6 shows the retreat of the glacier front and grounding line with time for the applied nine parameter combinations. The simulated evolution



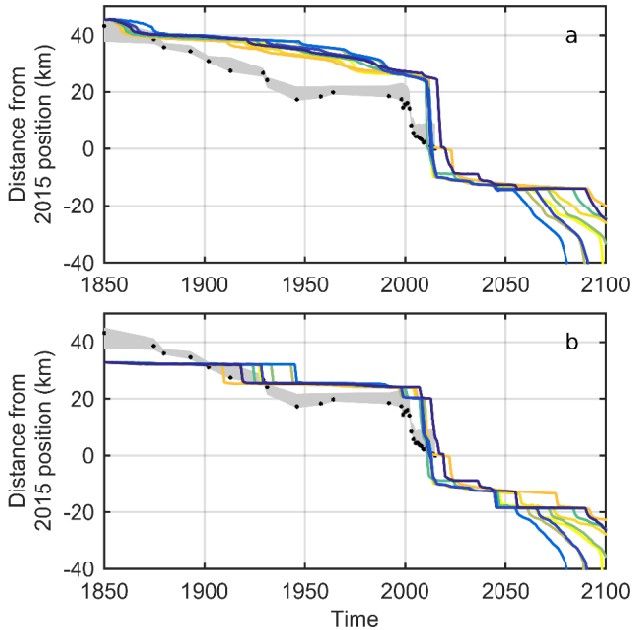

**Figure 6.** Simulated position of (a) the front and (b) the grounding line for nine different gradual forcing combinations. Colours correspond to the parameter combinations in Fig. 3. Black dots show the observed front positions at the centreline with a spread corresponding to the across-fjord variation of each front position (Fig. 1).

of the frontal position is similar for all experiments and shows the strong non-linearity even under a linear forcing (Fig. 6a). The different forcing experiments differ mainly in the timing of each further retreat, especially the retreat from the last stable position shown here just after 2050. The most markable event in the observations is the retreat of 19 km after the year 2000, which is simulated even more abruptly with all different parameter combinations as a retreat of at least 23 km. The simulated

frontal positions differ by up to 20 km from the observations, but due to the simplicity of the model, the aim is here to study the geometric controls on rapid retreat rather than tuning the model until the simulated retreat fits the observations.

Compared to the calving front, the grounding line retreats more step-wise (Fig. 6b). Before 2015, it stabilizes mainly at distances of 32 km, 25 km and 20 km from the 2015 frontal position for all experiments. The parameter combination thereby only determines the timing of the grounding line displacement to the next stable position. It retreats more gradually beyond

2015 with short stabilizations at 8 km, 12 km and 18 km upstream of the present-day position.

### 5.3  Control of fjord geometry on grounding line retreat

Figure 7 presents the stability of the grounding line for the different simplified geometries presented in Sect. 4.3. The stability is thereby quantified by how long the grounding line stays at one position. Figure 7a shows the original geometry with the most pronounced pinning points at a distance of 32 km and 25 km from the 2015 position as described in Sect. 5.2. Only the

length of stabilization thereby varies among the different parameter combinations, whereas the location is similar. Artificially





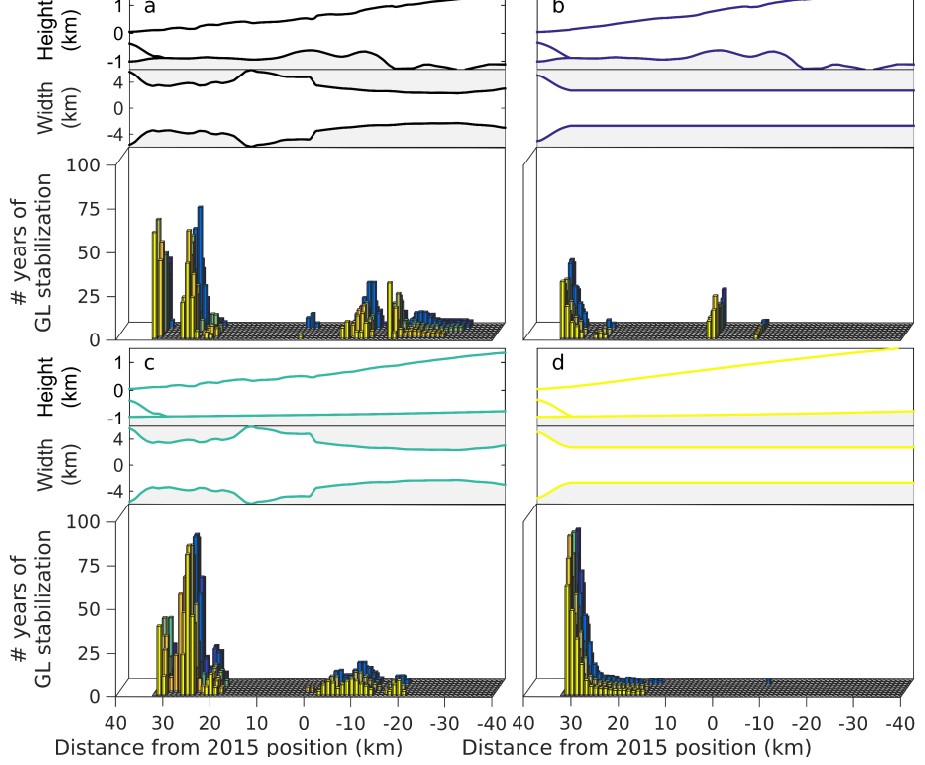

**Figure 7.** Stabilization of the grounding line (GL)0 for the different geometries presented in Sect. 4.1: (a) the original geometry, (b) straightened width, (c) straightened bed and (d) straightened width and bed. The bars represent the length of the stable period of the grounding line within 1 km (in years), and the colours correspond to the parameter combinations from Fig. 3. Only stable periods of more than two years are included.

straightening the width removes the pinning points at 25 km and those beyond the 2015 position. Instead, the glacier stabilizes at the present-day position and 10 km inland (Fig. 7b). Straightening the bed still results in similar pinning point locations at the lowermost 10 km compared to the original geometry, but spreads the stabilization beyond the 2015 position (Fig. 7c). Straightening the bed and the width removes all pinning points and leads to a linear response to the linear forcing. Note that all

5    geometries have an initial pinning point at the LIA position to permit a steady-state at the LIA position. Generally, a reduction in the complexity of the fjord geometry, e.g. straightening the bed and/or width reduces the number of pinning points.

## 6    Discussion

Our results highlight the importance of lateral and basal topography for the evolution of glacier retreat in fjords, JI on Greenland being only one example. The retreat of such glaciers is highly non-linear in response to a linear climate forcing. This challenges

10    our understanding of the recent observed retreat history and makes it hard to isolate the relative impact of changes in ocean



forcing, SMB and internal factors including the fjord geometry. Here, we discuss the impact of fjord geometry and compare the simulated glacier response to the recorded long term glacier retreat history, as well as explore the implications for the future response of JI to changes in climate.

## 6.1 Geometry more important than climate

Our simulations show that once a glacier retreat is triggered through changes at the marine boundary or at the glacier surface, a highly complex and non-linear response unfolds due to variations in the fjord geometry.

  For a retrograde bed, in a one-dimensional model, variations in the underlying bed topography influence the ice discharge, leading to an unstable glacial retreat (Weertman, 1974; Schoof, 2007). Previous studies also show that changes in the width of a glaciated fjord impact the lateral resistance, thereby stabilizing the glacier in narrow sections (Gudmundsson et al., 2012;

Jamieson et al., 2012). Figure 7 illustrates the enhanced non-linearity of the glacier response as the geometrical complexity increases. A flat glacier bed is less effective in reducing the non-linearity compared to straight lateral boundaries. However, it has to be considered that the glacier trough is an order of magnitude wider than it is deep, resulting in larger variations in the width compared to the bed. Therefore, we are not able to draw a firm conclusion on the relative importance of changes in the width versus the bed.

The simulations produce a rapid retreat of more than 20 km within one model year, given a small change in climate forcing. Although this one event is faster than what is observed for this time period shortly after the year 2000, it coincides with a wide section and a small depression in the bed, showing the importance of the fjord geometry in controlling the rate of retreat through time. In addition to the linear increase in climate forcing discussed in the results section, Fig. 8a presents the glacier response to a step increase in forcing at 1850, after which it is kept constant. The increase in the crevasse water depth,

submarine melt and reduction in sea ice buttressing is similar to the values reached at present-day for the linear increase and adjusted to fit the 2015 observed frontal position (we only present the best-fit experiment). With the step forcing, the glacier front remains relatively stable at a distance of 22 km, before it rapidly retreats to its 2015 position, where it is stable. The simulated retreat occurs within one year, despite a constant climate forcing. This unprovoked rapid retreat—after centuries of constant forcing—demonstrates the long response time of the glacier (Nye, 1960; Jóhannesson et al., 1989; Bamber et al.,

2007). The long response time is caused by a slow adjustment of the glacier volume to external changes. Figure 8b shows the change in glacier volume for the same experiments. For the step forcing, the volume adjusts steadily to the initial changes in forcing, despite the stable grounding line. During the rapid front retreat, the volume loss increases by 300 Gt and continues even after the grounding line stabilizes. This emphasizes that a stable grounding line does not imply that the glacier is in a steady-state. Similarly, an observed rapid retreat of a marine-terminating glacier might be the delayed response to past changes

in climate.

## 6.2 Predicting moraine positions

Figure 7 illustrates the potential in using the model simulations in a geomorphological context. Marine-terminating glaciers continuously erode their beds and deposit sediments, forming submarine landforms such as moraines. The rate of sediment





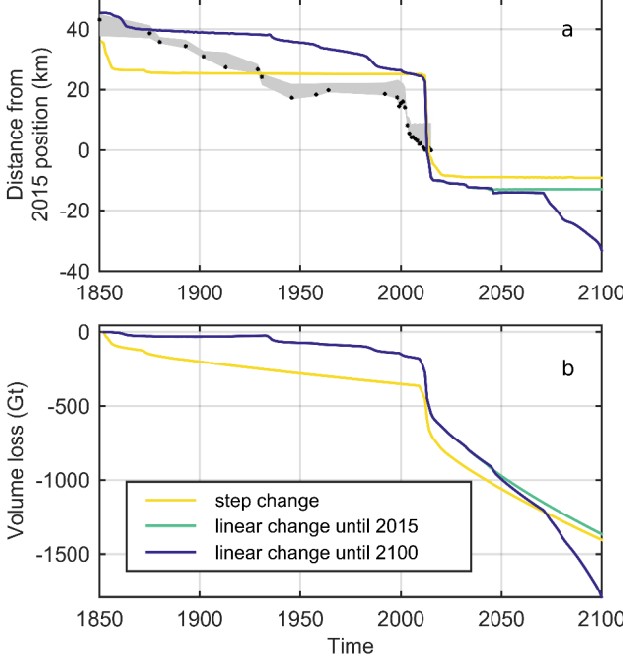

**Figure 8.** Simulated front positions (a) and accumulated volume loss (b) for different forcing scenarios. Black dots show the observed front positions at the centreline with a spread corresponding to the across-fjord variation of each front positions (Fig. 1).

deposition and resulting proglacial landforms are functions of climatic, geological and glaciological variables, though these functions remain poorly quantified due to sparse observational constraints. Proglacial transverse ridges tend to form during gradual grounded calving front retreat, whereas more pronounced grounding zone wedges are associated with episodic grounding line retreat (Dowdeswell et al., 2016).

5    The abundance of ice mélange in front of JI renders studies of submarine geomorphology difficult. Studies of this kind are lacking in the fjord, though evidence of the style of deglacial ice sheet retreat in Disko Bugt do exist (Streuff et al., 2017). However, our study raises more generic questions about the links between trough geometry and moraine positions. We suggest that moraine positions to a first order can be predicted from geometric information. In this context, the length of stabilisation in Fig. 7 can be viewed as a proxy for moraine build-up. Thereby, stable (moraine) positions are independent of model parameters, supporting geometric controls of moraine formation. This hypothesis remains to be tested with a proper model of sediment dynamics and constrained by a number of well-studied, diverse glaciological and climatic environments. While not a substitute for in-situ investigations, potential sites for more detailed (and costly) submarine studies could also be identified based on geometric information, using airborne or remotely sensed platforms. To this end, our study clearly highlights the potential of combing long-term modelling studies with geomorphological and sedimentary evidence to understand the non-linear response of marine ice sheet margins.





### 6.3 Limitations in simulation of glacier retreat history

We argue that fjord geometry to a large extent controls the retreat history of marine-terminating glaciers. Nevertheless, changes to the external forcing of the glacier are important as their magnitude times the onset of the retreat and determines its strength of the retreat (Fig. 6).

5    Although the observed rapid retreat after the disintegration of the floating tongue is simulated by the model, neither the step forcing nor the linear forcing reproduce all the details of the observed retreat history since the LIA (see Fig. 8a). However, a perfect match is not expected from a simple ice flow model as is used here, in particular given the linear forcing applied and the difficulty in measuring bed topography and bathymetry in this area (Boghosian et al., 2015).

Only certain parameter combinations—among others those presented in Fig. 3—attain a good match to the observed total 10    retreat history of JI since the LIA. If the submarine melt rate is increased, the crevasse water depth has to be reduced or/ and the sea ice buttressing increased, for modelled retreat to be in agreement with observations. Similarly, if the sea ice buttressing is reduced, the crevasse water depth and submarine melt rate have to be smaller (see Fig. 3). Importantly, none of the forcing parameters can trigger the retreat alone, given that they are perturbed within a reasonable range. As an example, individually, the submarine melt rate would have to be increased by 370 %, the crevasse water depth by 250 %, and the sea ice buttressing by 15    420 % to generate the required retreat. Although the increase in crevasse water depth may be reasonable, this increase leads to a water depth of 400 m in the crevasses, which is very high. These different thresholds for the forcing parameters show that the model is differently sensitive to the parameters; for instance a change in crevasse water depth of only a few % (depending on the absolute magnitude) delays the disintegration of the glacier tongue by several decades, whereas submarine melt rate has to be changed by an order of magnitude more (tens of %) to delay the disintegration by several decades. The strong sensitivity to 20    crevasse water depth results in the slightly different timing of the disintegration among the model runs shown in Fig. 6, which however does not distort our conclusions. With respect to absolute values for crevasse water depth and submarine melt rate, complexity of the physics behind theses processes and their implementation into numerical models has to be considered, as it leads to simplified parametrizations in this model. Values for the crevasse water depth are therefore exaggerated in our study to account for additional vertical submarine melt at the calving front, which is otherwise disregarded in the model.

25    Note that the SMB has an insignificant contribution to the frontal retreat, even if the frontal gradient is doubled and the SMB curve is lowered by 50 %, which gives a SMB of -6 m w.e. $\mathrm{yr}^{-1}$ at the terminus (compare with Fig. 2). In the model and for this glacier, changes in air temperatures therefore contribute mainly through runoff and the filling of crevasses with water, rather than directly through surface ablation. For the specific geometry of JI, the influx of ice at the lateral boundaries is a factor 100 larger than the SMB and could be important for the sensitivity of the glacier to changes in climate forcing. However, the lateral 30    flux has a minor impact on the retreat rate, and if changed solely, the lateral influx has to be decreased by nearly 70 % to match the observed retreat, which is deemed unrealistic.

In addition, the simulated break-off of the floating tongue is spatially and temporally overestimated, as observations show a retreat of 17 km within 12 years, whereas the model simulates a retreat of 13 km within one model year. However, in reality the glacier front was stabilized by a partly grounded floating tongue (Thomas et al., 2003). The stabilization of the grounding



line also caused a thickening and an advance by 3 km between 1991 and 1997 (Thomas et al., 2003), which is not resolved in the model. Indeed, pinning points that only reach partly across the fjord cannot be resolved in a width-averaged model. The simulated rapid disintegration of the floating tongue also leads to an overestimation of the velocities, giving higher ice discharge compared to observations. If assuming a smoother retreat of the grounding line, the modelled retreat gets closer to

the observations. Also the high uncertainty in calculating annual frontal positions from spatially highly variable and seasonally varying front positions has to be considered when comparing the modelled to observed retreat. For further studies that intend to reconstruct the observed retreat, or to project into the future, a good knowledge of the subglacial topography is required and the use of a three-dimensional model on a seasonal or even daily time-scale may be necessary.

### 6.4 Future of Jakobshavn Isbræ

Because of the long adjustment time of glaciers, model simulations require a long-term validation with data based on past climatic conditions, in order to understand recent as well as future behaviour. Initializing the simulations at the relatively stable position during the LIA maximum as done here, enables an about 150 year long validation with observations. With this validation, the steady-state glacier is forced with the right magnitude to reach a simulated retreat that compares relatively well to observations, considering the simplicity of the model. This allows for an extrapolation into the future, as shown in Fig. 6

and 8. For the geometry used and for all the applied forcing scenarios, the glacier front of JI continues to retreat rapidly inland until about 10 km beyond its position in 2015, where the retreat slows down (Fig. 5). For the experiment with linear increasing forcing, the glacier front retreats again after about 50 years, whereas it remains stable in the experiment with constant external forcing (Fig. 8). Due to the strong dependence of the grounding line stabilization on the subglacial and lateral topography, different fjord geometries can cause very different glacier retreat (Fig. 7). A good representation of the topography is therefore

crucial to attain accurate future projections. And as noted before, the long response time and memory of the glacier has to be considered, as one can expect dramatic changes in frontal positions decades after any significant changes in climate forcing.

### 7 Conclusions

The rapid retreat of many of Greenland's outlet glaciers during the last decades has been correlated with increased oceanic and atmospheric temperatures, though glaciers display diverse behaviour. We force a numerical model of JI from its LIA maximum

position in 1850 with a linear increase in SMB, submarine melt rate, crevasse water depth and reduction in sea ice buttressing to reproduce the observed retreat of 43.2 km since the LIA. The modelled LIA glacier has a 9 km long floating tongue and a velocity of 5 km yr$^{-1}$.

The observed retreat can be modelled by different combinations of linearly increased forcing parameters, whereas the SMB plays a negligible role. The response to the linear forcing is highly non-linear; we simulate a slow initial frontal retreat and

thinning, followed by the formation and rapid disintegration of a 16 km long floating tongue without additional forcing. The doubling in velocities caused by the collapse agrees well with observations.

We show that artificially straightening the bedrock and channel topography reduces the non-linearity of the glacier's retreat. Straightening removes geometric pinning points at which the grounding line stabilizes. Unstable positions occur on a reversed bed slope and at wider channel sections. We therefore argue that the retreat history since the LIA has largely been controlled by the geometry and that future retreat will be governed by similar factors. Since grounding line stability is fundamentally

controlled by the geometry, we also postulate that geometry can be used to infer sites of moraine formation.

While the geometry determines the positions of stabilization, external forcing controls the timing and pace of retreat. There is also an important distinction between temporary grounding line stabilization and steady-state, as the model glacier loses mass centuries after applied changes in external forcing. Together with unstable sections in the subglacial geometry, this can lead to rapid retreat, despite constant external forcing. In the future, JI may stabilize after a further retreat of 10 km, even for a

longer time in case of a constant climate. However, as the glacier has to adjust to the previous rapid retreat, it is possible that another rapid retreat follows, despite an unchanged external forcing.

*Code and data availability.* The model code is available through Faezeh M. Nick (faezeh.nick@gmail.com). The model output and other datasets can be obtained upon request from the corresponding author.

*Author contributions.* Nadine Steiger, Kerim H. Nisancioglu and Henning Åkesson designed the research, Nadine Steiger performed the

model runs and created the figures with significant input from Kerim H. Nisancioglu, Henning Åkesson and Basile de Fleurian. Faezeh Nick provided the model and technical support. Nadine Steiger wrote the paper, with substantial contributions from all authors.

*Competing interests.* The authors declare that they have no conflict of interest.

*Acknowledgements.* This research was funded by the Fast Track Initiative from Bjerknes Centre for Climate Research and the European Research Council under the European Community's Seventh Framework Programme (FP7/2007-2013)/ERC grant agreement 610055 as

part of the ice2ice project. Henning Åkesson was supported by the Research Council of Norway (project no. 229788/E10), as part of the research project Eurasian Ice Sheet and Climate Interactions (EISCLIM). Front positions of JI since 1990 are obtained from ENVO at http://products.esa-icesheets-cci.org/. We want to thank Mahé Perrette for providing a python-javascript project to produce a 1-D profile of bed topography and glacier surface, which is available at https://github.com/perrette/webglacier1d. Thanks also to Jason Box for providing SMB data for the GrIS.





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
