# Peer review of "Non-linear retreat of Jakobshavn Isbræ since the Little Ice Age controlled by geometry"

_The Cryosphere, 2017_

## Referee Comment (RC1) · J. H. Bondzio (Referee) · 11 Oct 2017

**Comments on N. Steiger et al., "Non-linear retreat of Jakobshavn Isbræ…" https://doi.org/10.5194/tc-2017-151**

J. H. Bondzio

October 10, 2017

**1  General Comments**

**1.1  Summary**

N. Steiger et al. set up a 1D flowline glacier model of Jakobshavn Isbræ (JI) and perform a sensitivity analysis on various climatic and geometric model input parameters for the glacier's evolution from the Little Ice Age (LIA) to today and into the future (until 2100). The authors conclude that the fjord and trough geometry are the main controls on the glacier's retreat history.

The question of what controls the retreat of JI and other marine-terminating glaciers in Greenland for the last decades is of great interest for assessments of the present and future mass balance of the Greenland ice sheet (GrIS). However, I find the present study has several shortcomings, which have to be addressed before the paper can be considered to be ready for publication. My main criticisms are: First, the 1D flowline model used for this study is inadequate to represent JI's complex flow dynamics, especially for inferring its grounding line position and for assessments of it's future evolution. Second, the results of the model study carry little to no novelty. Third, the manuscript would benefit of some shortening as well as a clearer structure. I will explain my criticism in more detail below.

**1.2  Model Fitness**

The 1D flowline model used for this study is inadequate to represent JI's complex flow dynamics. It is known that JI's flow regime is controlled by intense lateral shear in the shear margins, which has to be fully represented in any model of JI which aims to study its past and future evolution (e.g. Truffer and Echelmeyer, 2003; Joughin et al., 2012; Shapero et al., 2016; Bondzio et al., 2016). This 1D flowline model parametrizes the complex interaction of JI's fast-flowing trough and the surrounding inland ice, which is inadequate e.g. during rapid calving front retreat when large variations in ice viscosity and ice stream geometry

occur simultaneously in a non-linear manner (cf. e.g. Bondzio et al., 2017). The inadequacy shows e.g. in the overly rapid retreat of the calving front in the model.

Moreover, from this paper as it is, the parametrization of key physical processes like the lateral influx of ice into the ice stream as well as the grounding line remain unclear. I can not evaluate the appropriateness of their treatment as of now, and can therefore not evaluate whether the grounding line motion and glacier mass balance have been represented realistically.

Finally, these model shortcomings need to be mentioned in the manuscript and discussed as a limitation for the interpretation of the model results.

**1.3  Result Novelty**

While the sensitivity study by itself is an interesting model exercise, the results themselves lack novelty. It is known already that the fjord and bedrock geometry control the evolution of the glacier's calving front retreat and grounding line motion (Schoof, 2007; Morlighem et al., 2016), and this paper resembles in its setup strongly the study by Enderlin et al. (2013), who use the same model.

In my opinion, the argument of inference of moraine formation from glacier geometry is flawed. You argue that the bed geometry controls grounding line stabilization, and therefore the grounding line stabilization can be used to infer the bed geometry (i.e. moraine formation). Thus by knowing the bed we can infer the bed. This is a circular argument.

Moreover, the inadequacy of the model for JI, which does not allow to capture stable grounding lines on retrograde slopes, large errors in model input data near the grounding line, as well as the lacking description of the grounding line treatment leave me as of now sceptical towards any quantification of grounding line stabilization using this model, cf. point 2.54.

Finally, while it is tempting to produce projections of JI's future evolution, I believe that these projections are not reliable due to the above-mentioned model shortcomings, and similar projections produced using the same model have been presented elsewhere before (Nick et al., 2013).

**1.4  Structure**

The paper's structure should be presented more clearly. I recommend to adhere more strictly to the structure of theory, results and discussion. Some results and experiment setups are presented in the discussion for the first time, for example. For reasons of readability, I recommend to stick to the "1 paragraph, 1 message" structure, and start every paragraph with a sentence that states the paragraph's main message.

The paper is too long. The paper's main message is that geometry controls the glacier's retreat. Accordingly, only the information required to support this hypothesis should be included. Many observations listed in section 2 are not needed to support the results, and the model description in section 3 has already been given elsewhere (e.g. Enderlin et al., 2013). On the other hand, if

the authors wish to give an overview over existing observations on JI, then an overview over previous model studies should also be included.

The naming of model variables is sometimes inconsistent. The ice velocity is denoted sometimes with $U$, at other places with $v$, for example. Please include a complete table of model variables in the paper.

The experiments performed in this study should be described more clearly at one location in the paper only. For example, the results of a stepwise change in climate forcing is introduced only in the discussion. I suggest inserting a complete table of experiments, including all parameters and their values, in section 4.

Finally, the paper would benefit from a careful reread, as there are small grammatical errors and some misleading sentences. For more details, see the specific comments below.

**2 Specific Comments**

**2.1**

p1, abstract. I suggest shortening the abstract by summarizing the findings more.

**2.2**

The introduction carries too many details which are both widely known and not strictly needed for this study, and can therefore be dropped, e.g.:

- p1, l21 – p2, l2: The observations concerning the mass balance and flow regime of the GrIS.

- p2, l7: "Turbulent melt [. . . ]". Ocean melt processes are not explicitly modelled here.

**2.3**

p2, l14: "Notwithstanding widespread acceleration[. . . ]". Please rephrase.

**2.4**

p2, l19: "Here we therefore expand the range of climatic conditions[. . . ]". In this paper, you do not expand the range of climatic conditions, you use an expanded data set of climatic conditions reaching to the LIA in your model.

**2.5**

p2, l24: "The glacier's speed tripled within 20 years". The acceleration took place after 1998, giving a time span of only 14 years until 2012.

**2.6**

p2, l32: "landward sloping". Unclear formulation. I assume you mean landward down-sloping. In this case, the term "retrograde" is commonly used (as you do further down in the manuscript).

**2.7**

p2, l34: The cited studies make the findings that are stated in this study. Enderlin et al. (2013) explicitly studies the impact of fjord width on glacier geometry. The main findings of this study (geometry main control on retreat) are therefore not new. Moreover, Gudmundsson et al. (2012) finds that stable grounding lines on retrograde bedrocks in a deep trough are possible due to lateral stabilization. This geometric setting is exactly what defines JI, which is why 1D flowline models are inadequate for realistic modelling of JI.

**2.8**

p2,l35-p3,l2: Your result is that geometry controls retreat. In your literature overview you show that your findings are not new.

**2.9**

p3, l3-5: "Enderlin et al. (2013a) also showed that non-unique parameter combinations can exist for the same front positions, [...]" Perfect to be picked up in the discussion, as this corresponds to your findings as well.

**2.10**

p3, l5-6: "However, very limited knowledge exists (Lea et al., 2014; Jamieson et al., 2014) regarding the interplay between bedrock geometry, channel-width variations and external controls on a real glacier.": This interplay has been addressed previously by several studies, e.g. Enderlin et al. (2013); Morlighem et al. (2016). Your findings corroborate their studies, which should be mentioned in the discussion.

**2.11**

p3., l10: "non-linear frontal retreat". It is not possible to conclude from Bauer (1968) that the retreat prior to 1960 was non-linear or gradual, as the temporal sampling of the front positions is too sparse.

**2.12**

p3,l11: "43.2 km" The error on calving front positions, both from Bauer (1968) and given the seasonal variability in 2015, is larger than 0.1 km. The precision of the number given here and elsewhere in the manuscript is therefore too high.

**2.13**

p3, l12: "The aim of the study[...]". I recommend stating the study's aims clearly at the beginning of the paragraph in a positive formulation.

**2.14**

p3, l12: Model validation means checking the accuracy of the model's representation of the real system. Therefore, if you do not aim to represent JI as closely as possible, you can not validate your model. If validation of the model is not your aim, why don't you just perform your experiments on simplified geometries, as done earlier in Enderlin et al. (2013)?

**2.15**

p3, l14-22: This paragraph describes the content of the paper, and can be shortened in ways of: "Section 2 reviews the state of knowledge on JI's observations used for model validation, Section 3 describes the numerical ice flow model used here, Section 4 ..." etc.

**2.16**

p3, l24: "JI is the fastest and most active glacier on the GrIS (Legarsky and Gao, 2006),[...]". A better citation supporting your statement would be e.g. Rignot and Mouginot (2012).

**2.17**

p4, l24: "and has been studied extensively during the last decades". Please provide some key citations (e.g. publications by K. Echelmeyer, I. Joughin, M. Fahnestock, M. Truffer and others, as well as some modelling studies).

**2.18**

p3, l25-26: "We use this relatively well-observed glacier to analyse the controls of the geometry and external forcing on its rapid retreat since the LIA." Imprecise formulation: you use a numerical model and available observations to analyze the controls on the glacier's retreat.

**2.19**

Several observations are not necessary for this study, and should be dropped.

1. p3, l28: "inland of Disko Bugt". This is an unusual location description, and probably not necessary here, as the paper treats the ice sheet only.

2. p3, l30: "producing 10% of all icebergs released from the GrIS (Weidick and Bennike, 2007)." This observation is not used in this paper.

3. p4, l1-5: Ice margin positions prior to the LIA are not used in this paper.

4. p4, l9: "even". I am not sure why a re-advance of 3km in 1991 is worth mentioning given an annual front fluctuation of 2.5 km.

5. p4, l14: "Future simulations[...]" The future simulations are not used for discussion in this paper.

6. p5, l3: "using a combination of [...]" The technical details for the reconstruction are not contributing to this paper.

7. p5, l10: "Surface melting on the GrIS has increased by 63% ". Please use this to motivate your climate forcing choice, otherwise I'd drop it.

8. p5, l12: "due to a warming[...]". This part of the sentence is not re-used in the paper.

9. p6,l15: "The annual average in 2012[...]" This value is not used later on.

10. p6, l24: "The high discharge rates observed at JI may have already been initiated around 8 kyr BP, [...]". This observation is not used in the paper (you start in 1850).

**2.20**

p5, l6: "the upper area". Perhaps better: "at higher ice surface elevations"?

**2.21**

p5, l15: "two warming periods that are followed by a retreat of JI's calving front". Worth mentioning here also is the intermittent thinning of JI during these periods, stated by Csatho et al. (2008).

**2.22**

p6, l4: "seasonality of calving front migration;". At the end of this statement, several sources like Sohn et al. (1998); Joughin et al. (2004); Amundson et al. (2010) could be cited.

**2.23**

p6, l14: "Velocities display a strong seasonal cycle and [...]". This holds only for the time after the break-up, cf. Echelmeyer and Harrison (1990).

**2.24**

p6, l22: "most resistance". More precise: "most resistance to ice flow"

**2.25**

p6, l28: The array of observations identifies important processes for JI, which any model that is used for modelling JI's behavior has to capture. It would thus be useful to briefly state the most important process i.o. to motivate the model choice. Model shortcomings have to be stated clearly, and their implications for the discussion of results have to be clear.

**2.26**

p7, Sect. 3.1 & 3.2: The description of the ice flow model and calving parametrization has been given extensively in Enderlin et al. (2013), and can be replaced here using a reference to that paper. Only equation 6 could remain, as a new factor ($f_{si}$), has been added.

**2.27**

p7, l18: "The grounding line position [. . . ]". Please clarify what you mean by "robustly" and explain the grounding line treatment, which is a key process for the mechanics and results described in this paper (cf. e.g. section 6.2).

**2.28**

p8, table 1: The enhancement factor $E_{lat}$ is kept constant in time. Please discuss how this affects the results, as it is known that the ice viscosity in the shear margins drops significantly in response to glacier acceleration and calving front retreat (e.g. Bondzio et al., 2017).

**2.29**

Equations 3 & 4: Using a multiple-character symbol, e.g. $cwd$, for a variable is in conflict with standard notation in equations, where it is usually read as the product of three variables $c$, $w$, and $d$. Consider using different, one-symbol variables in the manuscript.

**2.30**

p9, l7: "In the model [. . . ] whole floating tongue (not shown here)". A conclusion is missing here. Which parametrization has been applied eventually? If only one has been used, the description of the other one can be dropped. Furthermore, which distance is used for the distance-dependent melting rate?

**2.31**

p9, l10-15: The physical motivation behind the lateral influx is unclear. First, what are the variables "velocity" $v_{L,0}$ and "depth" $H_{L,0}$? How and using which criterion have they been defined? How exactly do they change in time? Since

the lateral inflow is such an important component of the glacier's mass balance, and thus grounding line and calving behavior, these questions have to be answered clearly in the manuscript. Only then we can gauge whether they physical motivation behind the lateral inflow parametrization is sound.

**2.32**

p9, l13, 14 and Eq. 8: The ice velocity has been denoted by $U$ further above (Eq.1).

**2.33**

p9, Model setup: Please specify how you choose the centerline for your ice flow model.

**2.34**

p9, l22: "Bathymetry data and subglacial bed topography data for JI [. . . ]". For brevity reasons, please only describe the data sets you use. This sentence should be moved to section 2, or can be dropped altogether, as it does not contribute to the matter of the paper.

**2.35**

p10, l12: "Temperature profiles[. . . ]". Again, results from other studies do not need to be explicitly repeated in great detail, since for the purpose here we are only interested in the total temperature range. Moreover, ice temperatures at the ice divide are likely colder than -20 degrees Celsius, as even the present-day average annual surface temperature there is about -25 to -30 degrees Celsius (Ettema et al., 2009).

**2.36**

p10, l17: "Little change in ice temperature over the time scale[. . . ]". I believe this to be incorrect. Bondzio et al. (2017) show that the ice stream can indeed warm by several degrees Celsius during the flow acceleration following the disintegration of the ice tongue, which is significant for ice flow especially for the warm, soft ice near the terminus.

**2.37**

p10, l25: "linear increase". Please elaborate exactly how the submarine melt rate increases linearly. From when to when? From which value to which value?

**2.38**

p10, l27: "same gradual change in the SMB gradients from the LIA" is confusing to read as I they have been called "SMB profiles" in the caption of Fig. 2.

**2.39**

p10, l31: "The sea ice buttressing can be assumed to be linearly dependent on the ocean and air temperatures[. . . ]". Please cite the observational study that motivates this choice. The decrease in sea-ice cover by a factor three is not obvious as of now.

**2.40**

p10, l33: "However, a temperature increase [. . . ]". This parameter choice multiplies the longitudinal strain rate by up to a factor of 3, which will largely increase calving (cf. Eqs. 3-6). It should be discussed as of how realistic the results using such high values are.

**2.41**

p11, Fig. 3: It is hard to read the exact values of the parameters used here in this 3D plot. Using a grid in the back planes may help. The exponent in the unit of the submarine melt rate is off. However, instead of this figure, I recommend using a table which lists all parameters and their values, and which names the experiments.

**2.42**

p11, l1-4: The description of observations belongs into Section 2. Please include citations. Also, it is usually better to describe temperature changes in absolute numbers, as a "50%" increase in temperature is ambiguous for the various temperature units in use.

**2.43**

p11: The physical motivation of the parameter choices for submarine melt and crevasse water depth is poor. It would make much more sense to me if the author would simply say: "In order to perform a sensitivity analysis to different parameters, we vary parameter $X$ from . . . to . . . ". A sensitivity analysis should be in the center of every modeling study in order to gauge the robustness of the model results w.r.t. its input parameters (cf. also Enderlin et al., 2013).

**2.44**

p11, l7-11: "It is thereby tuned [. . . ]" From my understanding, figure 3 does not show how the parameters have been tuned to each other in order to achieve the

observed retreat. Neither does it show their interdependence, as it only shows the parameter space used in this sensitivity study. Please clarify this statement.

**2.45**

p12, l1: "well-known". This is an overstatement given the handful of calving front positions available from Bauer (1968) until the 1960s.

**2.46**

p12, l1-6: Please rephrase, the message of how the forcing perturbations are constrained is unclear. Furthermore, it is unclear how you obtain the calving front positions w.r.t. your flowline. Do you mean you take them as the intersection of the flow line used in the model with the observed calving front positions? Moreover, "latitudinal position" is an unsuitable coordinate, as JI's flow is not straight in the lowermost $\sim 60$ km.

**2.47**

p12, l22: "most reasonable". Please clarify.

**2.48**

p12, l28 to p13, l2: "The position of Camp-2 [. . . ]". It is hard to understand from this sentence how the position of Camp 2 on a surface bump leads to the overestimation of the ice thickness. Please clarify. Please explain which surface has been smoothed, ideally already in the model setup section.

**2.49**

p13, l7: "forced by [. . . ]". Please specify the parameters used for the forcing, as well as from when to when they have been increased linearly. In order to understand the results, the reader needs exact information on all forcings used in the model.

**2.50**

p14, figure 5c. Please clarify, are these annual average or maximum velocities? Please include the description of what is the grounding line flux (the circles?). Please add what is the color-coding of the circles. Please remove one of the y-axes to the right of plot c). Please use y-axis limits so that the circles are contained within the frame. Which locations from Joughin et al. (2014) are used for the velocities? Could you show them in a map?

**2.51**

p16, l8: "Our results highlight the importance [...]". This is no novel result, compare e.g. Morlighem et al. (2016); Enderlin et al. (2013), who should be cited here.

**2.52**

p17, l6: "a highly complex and non-linear response [...]". This response is linked to the variations in glacier width and depth, and therefore scale with their respective complexity, isn't it? I would formulate it like that.

**2.53**

Section 6.2 lacks a clear structure. It starts out by suggesting to use the model and fjord geometry to infer moraine positions, but then turns and wants to use moraine positions to understand the non-linear response of marine ice sheet margins. Both arguments remain unfinished, so please specify how you want to achieve either of your goals.

**2.54**

Moreover, while the geometric constrictions determine where the calving front stabilizes, there is little information on where the grounding line stabilizes, as this depends to a large degree on e.g. the ice stream's mass balance (i.e. lateral influx), the bedrock topography, and the treatment of the grounding line in the model (which remains unclear). Furthermore, as you state in your introduction, stable grounding lines on retrograde slopes are possible due to lateral stabilization (Gudmundsson et al., 2012). Last but not least, in order to answer whether moraines will form you need to know which process is dominant at a grounding line: substrate erosion or deposition. How can you tell that from your model? Given all the uncertainties in model input data and model parametrizations, please discuss how confident you can be in locations of grounding line stabilization obtained from your model, and hence moraine formation.

**2.55**

p18, l8: "can be predicted from geometric information[...]". You mentioned further above (p17, l13) that you cannot tell whether fjord width or depth is the dominant control on glacier retreat. Therefore, please specify which geometric information should be used to predict moraine formation, and how you can conclude that.

**2.56**

p19, l14: "sea ice buttressing". How can an increase in sea ice buttressing trigger retreat?

**2.57**

p19, l21-24: The message of these sentences is not clear to me. Please rephrase.

**2.58**

p19, Sect. 6.3. The discussion of which model parameter the glacier model is most sensitive to should happen in section 6.1. Instead, in section 6.3 you should discuss the limitations to the conclusions you are drawing, see comments above.

**2.59**

p19, l27: "SMB curve is lowered by 50 % [. . . ]". Lowering the SMB curve by 50% yields -4.5, not -6 m.w.e. $yr^{-}1$ at the terminus. Do you mean multiplying it by a factor two?

**2.60**

p20, l6: "For further studies [. . . ]". The reconstruction of the retreat has been done already by Bondzio et al. (2017), who should be cited here.

**2.61**

p20, Sect. 6.4 summarizes model results instead of discussing them. The results should be moved to the results section, and discussed here.

**3 Minor Corrections**

1. p2, l18: remain*s* challenging

2. p2, l18: "constrains" feels wrong choice here. Perhaps better inhibits?

3. p2, l19: Here, we [. . . ]

4. p2, l21: [. . . ] provides the context for [. . . ]

5. p2, l27: "largest sea level contribution". Better: largest contribution to sea level rise.

6. p3, l5: "very limited": A matter of taste, but the emphasis caused by "very" can be omitted here.

7. p3, l20: "Section 6," An excess comma.

8. p3, l29: "is drained". A matter of taste again, but using active form is more engaging for the reader.

9. p4, l13: "still ongoing retreat". Perhaps better: "which is ongoing as of today".

10. p5, l11: "is contemporary to". Better perhaps: "coincides with"?

11. p7, l4: "Ice thickness changes with time are calculated [...]". Please rephrase, "changes" reads as a verb, but is a noun, which is confusing to read.

12. p7, l7: "The mass balance B". B misses a dot.

13. p7, l24: "penetrates". Use plural here, as both surface crevasse depth and basal crevasse depth penetrate the glacier thickness.

14. p8, eq. 6: formatting: the dot over $\epsilon_{xx}$ spreads over to the subscript characters.

15. p9, l6: "relatively" can be dropped.

16. p12, l16: "straightened" is repeated three times here, please rephrase.

17. p12, l19: "geometry". Perhaps better: model setup?

18. p12, l26: "(Gudmundsson, 2003)". As you are presenting your results here, the citation can be dropped in my opinion.

19. p13, l5: "surface pumps", typo, do you mean surface bumps?

20. p13, l11: "the glacier terminus changes". Please clarify what characteristic of the terminus changes. I assume it's the configuration?

21. p17, l18: "In addition [...]" This starts a new topic, and deserves a new paragraph.

22. p19, l9: "among others those presented in Fig. 3". Please name them here for completeness. Also, discuss here earlier studies that found a similar result (e.g. Enderlin et al., 2013).

23. p19, l19: "order of magnitude". This formulation is usually used with respect to magnitude, and is confusing in this sentence, as it misleads the reader to believe that the melting rate has to be multiplied by a factor 10 or so in order to achieve what a few percent in the crevasse water depth. I believe that you want to say is that the submarine melt rate has to be changed by tens of percent to achieve an effect that is reached by changing the crevasse water depth by only a few percent, right? Please rephrase accordingly.

24. p21, l2: "Straightening". Please add: the bed.

**References**

Amundson, J. M., M. Fahnestock, M. Truffer, J. Brown, M. P. Lüthi, and R. J. Motyka
2010. Ice mélange dynamics and implications for terminus stability, Jakobshavn Isbræ, Greenland. *J. Geophys. Res. - Earth Surface*, 115(F1). F01005.

Bauer, A.
1968. Missions aeriennes de reconnaissance au Groenland 1957-1958; observations aeriennes et terrestres, exploitation des photographies aeriennes, determination des vitesses des glaciers velant dans Disko Bugt et Umanak Fjord. *Medd. om Grønland*, 173(3).

Bondzio, J. H., M. Morlighem, H. Seroussi, T. Kleiner, M. Rückamp, J. Mouginot, T. Moon, E. Y. Larour, and A. Humbert
2017. The mechanisms behind Jakobshavn Isbræ's acceleration and mass loss: A 3-D thermomechanical model study. *Geophys. Res. Lett.*, 44(12):6252–6260.

Bondzio, J. H., H. Seroussi, M. Morlighem, T. Kleiner, M. Rückamp, A. Humbert, and E. Larour
2016. Modelling calving front dynamics using a level-set method: application to Jakobshavn Isbræ, West Greenland. *The Cryosphere*, 10(2):497–510.

Csatho, B., T. Schenk, C. J. Van Der Veen, and W. B. Krabill
2008. Intermittent thinning of Jakobshavn Isbrae, West Greenland, since the Little Ice Age. *J. Glaciol.*, 54(184):131–144.

Echelmeyer, K. and W. Harrison
1990. Jakobshavns Isbræ, West Greenland - Seasonal variations in velocity or lack thereof. *J. Glaciol.*, 36(122):82–88.

Enderlin, E. M., I. M. Howat, and A. Vieli
2013. High sensitivity of tidewater outlet glacier dynamics to shape. *Cryosphere*, 7(3):1007–1015.

Ettema, J., M. R. van den Broeke, E. van Meijgaard, W. J. van de Berg, J. L. Bamber, J. E. Box, and R. C. Bales
2009. Higher surface mass balance of the Greenland Ice Sheet revealed by high-resolution climate modeling. *Geophys. Res. Lett.*, 36:1–5.

Gudmundsson, G. H., J. Krug, G. Durand, L. Favier, and O. Gagliardini
2012. The stability of grounding lines on retrograde slopes. *Cryosphere*, 6(6):1497–1505.

Joughin, I., W. Abdalati, and M. Fahnestock
2004. Large fluctuations in speed on Greenland's Jakobshavn Isbrae glacier. *Nature*, 432(7017):608–610.

Joughin, I., B. E. Smith, I. M. Howat, D. Floricioiu, R. B. Alley, M. Truffer, and M. Fahnestock
2012. Seasonal to decadal scale variations in the surface velocity of Jakobshavn Isbrae, Greenland: Observation and model-based analysis. *J. Geophys. Res.*, 117:1–20.

Joughin, I., B. E. Smith, D. E. Shean, and D. Floricioiu
2014. Brief Communication: Further summer speedup of Jakobshavn Isbræ. *The Cryosphere*, 8(1):209–214.

Morlighem, M., J. Bondzio, H. Seroussi, E. Rignot, E. Larour, A. Humbert, and S.-A. Rebuffi
2016. Modeling of Store Gletscher's calving dynamics, West Greenland, in response to ocean thermal forcing. *Geophys. Res. Lett.*, 43(6):2659–2666.

Nick, F. M., A. Vieli, M. L. Andersen, I. Joughin, A. Payne, T. L. Edwards, F. Pattyn, and R. S. W. van de Wal
2013. Future sea-level rise from Greenland's main outlet glaciers in a warming climate. *Nature*, 497(7448):235–238.

Rignot, E. and J. Mouginot
2012. Ice flow in Greenland for the International Polar Year 2008-2009. *Geophys. Res. Lett.*, 39, L11501:1–7.

Schoof, C.
2007. Ice sheet grounding line dynamics: Steady states, stability, and hysteresis. *J. Geophys. Res.*, 112(F03S28):1–19.

Shapero, D. R., I. R. Joughin, K. Poinar, M. Morlighem, and F. Gillet-Chaulet
2016. Basal resistance for three of the largest Greenland outlet glaciers. *J. Geophys. Res.*, 121(1):168–180. 2015JF003643.

Sohn, H., K. Jezek, and C. van der Veen
1998. Jakobshavn Glacier, West Greenland: 30 years of spaceborne observations. *Geophys. Res. Lett.*, 25(14):2699–2702.

Truffer, M. and K. Echelmeyer
2003. Of isbræ and ice streams. *Ann. Glaciol.*, 36:66–72. International Symposium on Fast Glacier Flow, Yakutat, Alaska, Jun 10-14, 2002.

---

## Author Comment (AC1) · 13 Nov 2017

Reply to comments from October 10, 2017 by J.H. Bondzio
on N. Steiger et al., "Non-linear retreat of Jakobshavn Isbræ. . . "

https://doi.org/10.5194/tc-2017-151

November 13, 2017

**1 General Comments**

**1.1 Summary**

N. Steiger et al. set up a 1D flowline glacier model of Jakobshavn Isbræ (JI) and perform a sensitivity analysis on various climatic and geometric model input parameters for the glacier's evolution from the Little Ice Age (LIA) to today and into the future (until 2100). The authors conclude that the fjord and trough geometry are the main controls on the glacier's retreat history.

The question of what controls the retreat of JI and other marine-terminating glaciers in Greenland for the last decades is of great interest for assessments of the present and future mass balance of the Greenland ice sheet (GrIS). However, I find the present study has several shortcomings, which have to be addressed before the paper can be considered to be ready for publication. My main criticisms are: First, the 1D flowline model used for this study is inadequate to represent JI's complex flow dynamics, especially for inferring its grounding line position and for assessments of it's future evolution. Second, the results of the model study carry little to no novelty. Third, the manuscript would benefit of some shortening as well as a clearer structure. I will explain my criticism in more detail below.

*We wish to thank the first reviewer for his comments and will address the three main concerns raised here together with the remaining detailed comments in the following.*

*An outline of how we will rewrite the manuscript to address the suggestions in the review is given in the following. Once we receive all reviews we will provide an updated version of the manuscript.*

**1.2 Model Fitness**

The 1D flowline model used for this study is inadequate to represent JI's complex flow dynamics. It is known that JI's flow regime is controlled by intense lateral shear in the shear margins, which has to be fully represented in any model of JI which aims to study its past and future evolution (e.g. Truffer and Echelmeyer, 2003; Joughin et al., 2012; Shapero et al., 2016; Bondzio et al., 2016).

*We are very much aware of the fact that JI has complex flow dynamics and that our study applies an idealized flowline model that does not capture all aspects of the dynamics.*

*However, the aim of our study is to use JI (which has a relatively well documented retreat history) as a test case to investigate the relative impact of fjord geometry (in particular fjord depth and width) to climate forcing, rather than tuning a model to reconstruct the history as realistically as possible. It is clear from our study and previous studies cited, but not necessarily in the scientific community at large, that fjord geometry dominates the long term retreat history of JI (over the climate forcing). The implications of this is that we might be able to predict the long term response of JI given detailed knowledge of the underlying upstream bed topography and fjord width.*

*Reviewing the literature, there is no clear consensus on the strength of the lateral and basal shear at JI, especially in a width-averaged setting. Some studies find that the base in the trough is very slippery, explaining the high velocities (Luthi et al., 2002; Shapero et al., 2016, Bondzio et al., 2017). However, these are observations at only one distance from the terminus (Luthi et al) and an inverse model that does not include any weakening at the lateral shear margins through crevassing and meltwater penetration (Shapero et al), which is suggested as main reason for the high velocities elsewhere (Van der Veen, 2011). Joughin et al. (2012) only find a very weak bed in the trough using a stiff model, which is in disagreement with the observations, whereas they find a less pronounced difference between the basal resistance in the trough and the margins, when using a soft model. We agree on that models that are used to project the future of JI should be more complex and include a full representation of the lateral shear. However, as we aim to study the effect of trough geometry over a long time period (in which detailed observations are missing to force a complex model) rather than projecting the future, we chose a flowline due to efficiency and a robust treatment of the grounding line and explicit representation of the calving front. In this model, lateral shear is parametrized by definition. However, we use a lateral enhancement factor of 10 to account for strong lateral resistance and will discuss the implications of the parametrization in the revised manuscript.*

This 1D flowline model parametrizes the complex interaction of JI's fast-flowing trough and the surrounding inland ice, which is inadequate e.g. during rapid calving front retreat when large variations in ice viscosity and ice stream geometry occur simultaneously in a non-linear manner (cf. e.g. Bondzio et al., 2017). The inadequacy shows e.g. in the overly rapid retreat of the calving front in the model.

*Deviation of the modelled retreat from observations are discussed in the paper (see Section 6.3, l.5 and l.32) as a result of the linear forcing, the simplicity of the model, and the lack of knowledge/ width-averaging of the bed topography. We do not believe that the overly rapid retreat is solely a consequence of the parametrization of ice viscosity, but rather a consequence of the width-averaging not capturing a partly grounded floating tongue (discussed in Section 6.3, l.32; Thomas et al., 2003). Note that the effective viscosity in our model is calculated in each time step with a non-linear dependency on the strain rate (see e.g. Nick et al., 2010).  However, we agree that a more dynamic ice viscosity and the interplay with ice stream geometry is important, at least for models used for projections. We will discuss this limitation in more detail in the revised manuscript.*

*Note that the 1D flowline model does have shortcomings in its representation of ice dynamics, as is noted in our manuscript, however, it also has its advantages. In particular, it is extremely efficient, making it possible to run a very large number of ensemble experiments testing the impact of different forcing factors and choices of parameter values (as an example more than 2000 simulations (only the relevant once shown in the paper) of 250 years were completed during this study). In addition to this, the model includes an explicit, physical treatment of calving and frontal dynamics which allows for the study of transient frontal retreat given changes in the climate forcing (submarinemelt, sea ice buttressing, surface melt and crevasse water depth). In many studies, including those with 3D dynamical flow models, the frontal position through time is not explicitly resolved, and is rather fixed in time or forced to a particular position manually (e.g. Gillet-Chaulet et al., 2012; Cornford et al., 2015; Bondzio et al., 2017). The latter would not have been possible in our study as this is one of the main points of interest – namely the transient response of the glacier front given changes in climate and fjord geometry.*

Moreover, from this paper as it is, the parametrization of key physical processes like the lateral influx of ice into the ice stream as well as the grounding line remain unclear. I can not evaluate the

appropriateness of their treatment as of now, and can therefore not evaluate whether the grounding line motion and glacier mass balance have been represented realistically.

*Thanks for pointing this out. The parametrization of the lateral influx of ice is explained in section 3.3. and we will improve the corresponding paragraph: The lateral influx Q_L is initially calculated at each grid point as the sum of the northern and southern lateral fluxes, given by observed velocities and thickness (Rignot and Mouginot, 2012; Morlighem et al., 2014) weighted with the width of the main trough. As we assume that the influx evolves with time, we scale it with the change in ice flux through the main tough.*

*Initial lateral flux:* $\quad Q_{L,0} = \dfrac{v_{L,0}(x) \cdot H_{L,0}(x)}{W_{JI}(x)}$

*Lateral flux with time:* $\quad Q_{L,t} = Q_{L,0}(x) \cdot \dfrac{v_{JI,t}(x) \cdot H_{JI,t}(x)}{v_{JI,0}(x) \cdot H_{JI,0}(x)}$

*(index JI = main trough of JI; index L = velocities and thickness of the inflowing ice)*

*The grounding line position is calculated for each time step with a flotation criterion based on van der Veen (1996), in which the ice is floating when the ice thickness is less than the flotation thickness. The glacier front position is calculated with the fully dynamic crevasse-depth calving criterion (Nick et al., 2010; see Section 3.2 in our paper), which calculates calving where surface and basal crevasses penetrate the whole glacier thickness. This enables simulation of transitions from a grounded front to a floating glacier tongue, and vice versa, and is linked to climate through the calculation of crevasse depths by penetration of surface melt water. The moving spatial grid adjusts freely to the new glacier length in each time step, tracking the glacier grounding line continuously based on hydrostasy (Vieli and Payne, 2005; Nick et al., 2009, 2010). This allows for a precise simulation of the glacier front and grounding line position using very high grid resolution. The grid size is set to 300 m initially, which decreases further as the glacier retreats and the length decreases. We will include a more detailed description of the grounding line dynamics in the revised manuscript.*

Finally, these model shortcomings need to be mentioned in the manuscript and discussed as a limitation for the interpretation of the model results.

*We will include a discussion of the limitations of the ice flow model in section 6.3 "Limitations in simulation of glacier retreat history", where we discuss the model and its difference to 3D models, and how the simplifications applied might impact the results.*

**1.3 Result Novelty**

While the sensitivity study by itself is an interesting model exercise, the results themselves lack novelty. It is known already that the fjord and bedrock geometry control the evolution of the glacier's calving front retreat and grounding line motion (Schoof, 2007; Morlighem et al., 2016), and this paper resembles in its setup strongly the study by Enderlin et al. (2013), who use the same model.

*There are indeed other studies showing the importance of bedrock geometry. However, in the community there is a strong emphasis on the role of ice-ocean interactions as a key control on the retreat of marine terminating glaciers such as JI (e.g. Holland et al., 2008; Joughin et al., 2012; Straneo and Heimbach, 2013; Cook et al., 2016; Mengel et al., 2016). In particular, it is often implied that changes in climate, ocean circulation and temperature are the main controls on glacier retreat.*

*This is only partly true, as once a retreat at a marine margin is triggered, the influence of bedrock geometry and fjord width dominate the transient response of the glacier, as is shown in our study as well as others (e.g. Jamieson et al. 2012; 2013; Morlighem et al. 2016).*

*In addition, , most studies only consider the recent retreat of glaciers such as JI and not the long term response where fjord geometry becomes more important for the transient response (e.g. Nick et al., 2013; Muresan et al., 2015; Bondzio et al., 2017). In this study we consider the period starting at the Little Ice Age which clearly is novel for JI and gives a significant increase in the data available to study the long term response of marine terminating glaciers such as JI. We would claim that only using the recent observed retreat might be misleading and is not adequate for understanding the long term retreat of marine terminating glaciers in Greenland, in particular, because fjord geometry is such an important factor on longer timescales. In essence, past changes in climate could be more important than contemporary changes in triggering the recent observed retreat given timescales involved. Similarly, the current changes in climate may trigger a delayed rapid retreat in the future of presently stable glaciers.*

*However, perhaps the most novel finding of our study is that an understanding of the relative impact of fjord geometry can help predict the position of past moraine positions in fjord systems such as Jakobshavn. This information will be particularly valuable in assessing the geometry and glacial history of similar fjord systems such as those in Greenland, Norway, Patagonia, and in Alaska.*

*To make the novelty of our study clear to the reader we will further emphasize these points in the revised manuscript, as well as discuss our study in light of the references mentioned, as well as the recent publication on JI by the reviewer (which was not available at the time of submission of this manuscript to TCD).*

In my opinion, the argument of inference of moraine formation from glacier geometry is flawed. You argue that the bed geometry controls grounding line stabilization, and therefore the grounding line stabilization can be used to infer the bed geometry (i.e. moraine formation). Thus by knowing the bed we can infer the bed. This is a circular argument.

*We apologize for the misunderstanding. Our argument refers both on the dependency of grounding line stabilization on bed geometry and trough width (see Figure 7), and how likely positions of moraines can be inferred from the fjord geometry. We will make this clear in the revised manuscript.*

*Note that due to the stabilization of the grounding line on inland up-sloping bedrock features, moraines may also be formed on top of bedrock bumps. Measurements of the bedrock in fjords like Jakobshavn are difficult to conduct, as gravity-derived measurements rely on knowledge of the density of the bed, which is dependent on the sediment thickness and the underlying bedrock geology (Boghosian et al., 2015). We that using the simulated duration of stabilization of the grounding line through time (figure 7) as a proxy for moraine build-up could help to provide information on the likely positions of moraines in the fjord.*

*We will make this clear in the revised manuscript.*

Moreover, the inadequacy of the model for JI, which does not allow to capture stable grounding lines on retrograde slopes, large errors in model input data near the grounding line, as well as the lacking description of the grounding line treatment leave me as of now sceptical towards any quantification of grounding line stabilization using this model, cf. Point 2.54.

*The model used does capture stable grounding lines on retrograde slopes, provided there is a narrowing in trough width as shown by Jamieson et al. (2012), who used a similar model. Regarding the lack of description, see above.*

Finally, while it is tempting to produce projections of JI's future evolution, I believe that these projections are not reliable due to the above-mentioned model shortcomings, and similar projections produced using the same model have been presented elsewhere before (Nick et al., 2013).

*We do not intend to project JI's future evolution as this has already been done – as you commented. Therefore, we wrote that it allows an "extrapolation into the future" (p. 20, l.14), which is rather done to increase the timespan and range of geometric variability, with the aim to study the non-linearity of the glacier retreat related to the geometry. We will improve section 6.4. and clarify that this is not meant as a projection on JI's future.*

*Note, however, that our study carries important implications for future modelling, such as the importance of including a long time span (e.g. since the Little Ice Age) to account for delayed responses to previous changes in forcing conditions. Previous studies that present projections, in contrast, only had a limited temporal range of observations to test the response of the model and to test its performance before attempting to simulate further retreat upstream.*

**1.4 Structure**

The paper's structure should be presented more clearly. I recommend to adhere more strictly to the structure of theory, results and discussion. Some results and experiment setups are presented in the discussion for the first time, for example. For reasons of readability, I recommend to stick to the "1 paragraph, 1 message" structure, and start every paragraph with a sentence that states the paragraph's main message.

*Thanks for the advise. We will improve the readability of our manuscript.*

The paper is too long. The paper's main message is that geometry controls the glacier's retreat. Accordingly, only the information required to support this hypothesis should be included. Many observations listed in section 2 are not needed to support the results, and the model description in section 3 has already been given elsewhere (e.g. Enderlin et al., 2013). On the other hand, if the authors wish to give an overview over existing observations on JI, then an overview over previous model studies should also be included.

*We will shorten section 2 on the background of Jakobshavn Isbræ and rather include references for the model description. We will also include more details of recent studies in the discussions (in particular the reviewer's recent paper on JI, Bondzio et al. 2017).*

The naming of model variables is sometimes inconsistent. The ice velocity is denoted sometimes with $U$, at other places with $v$, for example. Please include a complete table of model variables in the paper.

*Thanks. This will be included in revised manuscript.*

The experiments performed in this study should be described more clearly at one location in the paper only. For example, the results of a stepwise change in climate forcing is introduced only in the discussion. I suggest inserting a complete table of experiments, including all parameters and their values, in section 4.

*The stepwise change is only used as an example in the discussion and was not considered one of the main results, which is why we wanted to keep it separate. However, we will restructure this part for*

*better readability.*

Finally, the paper would benefit from a careful reread, as there are small grammatical errors and some misleading sentences. For more details, see the specific comments below.

*Thanks for many constructive specific comments on the text, see our detailed reply below. All comments will be included in revised manuscript.*

**2 Specific Comments**

2.1

p1, abstract. I suggest shortening the abstract by summarizing the findings more.

*We will rewrite the abstract with a stronger focus on our main findings, after we received the other review(s).*

2.2

The introduction carries too many details which are both widely known and not strictly needed for this study, and can therefore be dropped, e.g.:

• p1, l21 – p2, l2: The observations concerning the mass balance and flow regime of the GrIS.

*Okay, we can shorten this:*
*Marine-terminating glaciers export ice from the interior of the Greenland Ice Sheet (GrIS) through deep troughs terminating in fjords. Dynamic discharge accounts for about half of the current GrIS mass loss (Khan et al., 2015) and is impacted by several processes linked to air and ocean temperatures...*

• p2, l7: "Turbulent melt [. . . ]". Ocean melt processes are not explicitly modelled here.

*Agree, drop sentence.*

2.3

p2, l14: "Notwithstanding widespread acceleration[. . . ]". Please rephrase.

*Despite widespread acceleration...*

2.4

p2, l19: "Here we therefore expand the range of climatic conditions[. . . ]". In this paper, you do not expand the range of climatic conditions, you use an expanded data set of climatic conditions reaching to the LIA in your model.

*Here we therefore use an expanded data set of climatic conditions reaching from the Little Ice Age (LIA) maximum in 1850 to present-day.*

2.5

p2, l24: "The glacier's speed tripled within 20 years". The acceleration took place after 1998, giving a time span of only 14 years until 2012.

*Well, comparing the numbers in 1992 and 2012, it gives a speedup by 286% (annual average) – 420% (summer value) (see Joughin 2014).*

2.6

p2, l32: "landward sloping". Unclear formulation. I assume you mean landward down-sloping. In this case, the term "retrograde" is commonly used (as you do further down in the manuscript).

*Thanks, will rename*

2.7

p2, l34: The cited studies make the findings that are stated in this study.

Enderlin et al. (2013) explicitly studies the impact of fjord width on glacier geometry. The main findings of this study (geometry main control on retreat) are therefore not new. Moreover, Gudmundsson et al. (2012) finds that stable grounding lines on retrograde bedrocks in a deep trough are possible due to lateral stabilization. This geometric setting is exactly what defines JI, which is why 1D flowline models are inadequate for realistic modelling of JI.

*As stated in comment 1.3, we are awarethat there are other studies (as e.g. Enderlin et al and Gudmundsson et al) showing the importance of bedrock geometry. However, our application on a real glacier system allows for a validation of the findings with observations and we expand on a much longer time scale than most studies do. Also, we propose important implications such as the delayed rapid retreat as a response to preceded climatic changes or the build-up of moraines at geometric pinning points. This communicates the importance of the fjord geometry to readers beyond glacier modellers and promotes closer futurecollaboration between geomorphologists and modellers.*

2.8

p2,l35-p3,l2: Your result is that geometry controls retreat. In your literature overview you show that your findings are not new.

*See 2.7*

2.9

p3, l3-5: "Enderlin et al. (2013a) also showed that non-unique parameter combinations can exist for the same front positions, [...]" Perfect to be picked up in the discussion, as this corresponds to your findings as well.

*Thanks for the suggestions. We will pick up the possibility of same front positions with different parameter combinations in the discussion.*

2.10

p3, l5-6: "However, very limited knowledge exists (Lea et al., 2014; Jamieson et al., 2014) regarding the interplay between bedrock geometry, channel-width variations and external controls on a real glacier.": This interplay has been addressed previously by several studies, e.g. Enderlin et al. (2013); Morlighem et al. (2016). Your findings corroborate their studies, which should be mentioned in the discussion.

*We will include previous studies that show similar findings in our discussion section 6.1 Geometry more important than climate.*

2.11

p3., l10: "non-linear frontal retreat". It is not possible to conclude from Bauer (1968) that the retreat prior to 1960 was non-linear or gradual, as the temporal sampling of the front positions is too sparse.

*We agree on that there were for sure some linear periods in-between, and we don't know much about the first period. But we are here considering the whole time period, in which the retreat was for sure non-linear (see Figure 6) and other literature mentioned in p3. L10.*

2.12

p3,l11: "43.2 km" The error on calving front positions, both from Bauer (1968) and given the seasonal variability in 2015, is larger than 0.1 km. The precision of the number given here and elsewhere in the manuscript is therefore too high.

*Okay, good point.*

2.13

p3, l12: "The aim of the study[. . . ]". I recommend stating the study's aims clearly at the beginning of the paragraph in a positive formulation.

*Thanks, we will rewrite the sentence in the revised manuscript.*

2.14

p3, l12: Model validation means checking the accuracy of the model's representation of the real system. Therefore, if you do not aim to represent JI as closely as possible, you can not validate your model. If validation of the model is not your aim, why don't you just perform your experiments on simplified geometries, as done earlier in Enderlin et al. (2013)?

*See 2.13*

2.15

p3, l14-22: This paragraph describes the content of the paper, and can be shortened in ways of: "Section 2 reviews the state of knowledge on JI's observations used for model validation, Section 3 describes the numerical ice flow model used here, Section 4 . . . " etc.

*We will shorten the paragraph in the revised manuscript.*

2.16

p3, l24: "JI is the fastest and most active glacier on the GrIS (Legarsky and Gao, 2006),[. . . ]". A better citation supporting your statement would be e.g. Rignot and Mouginot (2012).

*Okay, thanks.*

2.17

p4, l24: "and has been studied extensively during the last decades". Please provide some key citations (e.g. publications by K. Echelmeyer, I. Joughin, M., Fahnestock, M. Truffer and others, as well as some modelling studies).

*Yes, sure.*

2.18

p3, l25-26: "We use this relatively well-observed glacier to analyse the controls of the geometry and external forcing on its rapid retreat since the LIA." Imprecise formulation: you use a numerical model and available observations to analyze the controls on the glacier's retreat.

*Yes, thanks.*

2.19

Several observations are not necessary for this study, and should be dropped.

1. p3, l28: "inland of Disko Bugt". This is an unusual location description,

and probably not necessary here, as the paper treats the ice sheet only. *Okay.*

2. p3, l30: "producing 10% of all icebergs released from the GrIS (Weidick

and Bennike, 2007)." This observation is not used in this paper. *Okay, dropped*

3. p4, l1-5: Ice margin positions prior to the LIA are not used in this paper.

*No, but the link between grounding line stabilization and moraine build-up is used. See the improved suggestion below.*

4. p4, l9: "even". I am not sure why a re-advance of 3km in 1991 is worth

mentioning given an annual front fluctuation of 2.5 km.

5. p4, l14: "Future simulations[. . . ]" The future simulations are not used for discussion in this paper. *Will be included.*

6. p5, l3: "using a combination of [. . . ]" The technical details for the reconstruction are not contributing to this paper.

*Okay, being dropped.*

7. p5, l10: "Surface melting on the GrIS has increased by 63% ". Please use

this to motivate your climate forcing choice, otherwise I'd drop it.

*We use it for the crevasse water depth, see p.11 l. 5*

8. p5, l12: "due to a warming[. . . ]". This part of the sentence is not re-used in the paper.
*Okay, can be dropped*

9. p6,l15: "The annual average in 2012[. . . ]" This value is not used later on.
*Yes, it is. See Fig. 5.*

10. p6, l24: "The high discharge rates observed at JI may have already been

initiated around 8 kyr BP, [. . . ]". This observation is not used in the

paper (you start in 1850). *Okay, will be dropped*

2.20

p5, l6: "the upper area". Perhaps better: "at higher ice surface elevations"?   *Okay*

2.21

p5, l15: "two warming periods that are followed by a retreat of JI's calving front". Worth mentioning here also is the intermittent thinning of JI during these periods, stated by Csatho et al. (2008). *Yes, we will mention it.*

2.22

p6, l4: "seasonality of calving front migration;". At the end of this statement, several sources like Sohn et al. (1998); Joughin et al. (2004); Amundson et al. (2010) could be cited.

*Thanks for providing important references.*

2.23

p6, l14: "Velocities display a strong seasonal cycle and [. . . ]". This holds only for the time after the break-up, cf. Echelmeyer and Harrison (1990).

*Interesting point! We will add the citation and mention that velocities display a strong seasonal cycle after the break-up.*

2.24

p6, l22: "most resistance". More precise: "most resistance to ice flow" *Okay*

2.25

p6, l28: The array of observations identifies important processes for JI, which any model that is used for modelling JI's behavior has to capture. It would thus be useful to briefly state the most important process i.o. to motivate the model choice. Model shortcomings have to be stated clearly, and their implications for the discussion of results have to be clear.

*As suggested in 2.19, we will drop some of the observations that are not used for the study. The motivations for the model choice are the efficiency that allows to include a time span from the Little Ice Age, which we think is crucial, but also the explicit physical treatment of the grounding line and the use of a calving criterion to calculate the front positions.*

2.26

p7, Sect. 3.1 & 3.2: The description of the ice flow model and calving parametrization has been given extensively in Enderlin et al. (2013), and can be replaced here using a reference to that paper. Only equation 6 could remain, as a new factor (fsi), has been added.

*For the revised manuscript, we will consider to remove some equations and rather refer to Nick et al. (2010), which is the main reference to our model.*

2.27

p7, l18: "The grounding line position [. . . ]". Please clarify what you mean by "robustly" and explain the grounding line treatment, which is a key process for the mechanics and results described in this paper (cf. e.g. section 6.2).

*We will include a more detailed description of the grounding line treatment, see our reply to comment 1.2*

2.28

p8, table 1: The enhancement factor Elat is kept constant in time. Please discuss how this affects the results, as it is known that the ice viscosity in the shear margins drops significantly in response to glacier acceleration and calving front retreat (e.g. Bondzio et al., 2017).

*The stated reference was not published yet at the time of our submission. We will refer to it in the revised manuscript and discuss the implications of using a constant enhancement factor.*

2.29

Equations 3 & 4: Using a multiple-character symbol, e.g. cwd, for a variable is in conflict with standard notation in equations, where it is usually read as the product of three variables c, w, and d. Consider using different, one-symbol variables in the manuscript.

*Okay, we will rename the variable.*

2.30

p9, l7: "In the model [. . . ] whole floating tongue (not shown here)". A conclusion is missing here. Which parametrization has been applied eventually?

If only one has been used, the description of the other one can be dropped.

Furthermore, which distance is used for the distance-dependent melting rate?

*We will clarify that a constant value  for submarine melt is used eventually, which is justified by stating that we also used a distance-depend melting rate that gave results insignificantly different to the results with a constant value. The distance is the distance between the calving front and the grounding line.*

2.31

p9, l10-15: The physical motivation behind the lateral influx is unclear. First, what are the variables "velocity" $v_{L,0}$ and "depth" $H_{L,0}$? How and using which criterion have they been defined? How exactly do they change in time? Sincethe lateral inflow is such an important component of the glacier's mass balance, and thus grounding line and calving behavior, these questions have to be answered clearly in the manuscript. Only then we can gauge whether they physical motivation behind the lateral inflow parametrization is sound.

*The implementation of the lateral fluxes is inspired by Lea et al. (2014) and based on mass conservation. Due to the one dimensionality of the model, the influx at the lateral margins ($Q = v*H$) has to be divided by the width of the main channel at each grid point. We here assume that the lateral influx changes velocity and thickness in a similar rate as the main trough and therefore scale the original flux with the flux change of the main trough. See also our response to 1.2. We will include a better description of the parametrization of the inflow in the revised manuscript.*

2.32

p9, l13, 14 and Eq. 8: The ice velocity has been denoted by U further above (Eq.1). *Okay, we will change this.*

2.33

p9, Model setup: Please specify how you choose the centerline for your ice flow model. *It is explained on p.11, l.6 "We first calculate a centreline as a smoothed line following the mean latitudinal position of each observed glacier front", but we can include it in the model setup.*

2.34

p9, l22: "Bathymetry data and subglacial bed topography data for JI [. . . ]". For brevity reasons, please only describe the data sets you use. This sentence should be moved to section 2, or can be dropped altogether, as it does not contribute to the matter of the paper.
*Okay, we remove observations that are not used in our study.*

2.35

p10, l12: "Temperature profiles[. . . ]". Again, results from other studies do not need to be explicitly repeated in great detail, since for the purpose here we are only interested in the total temperature range. Moreover, ice temperatures at the ice divide are likely colder than -20 degrees Celsius, as even the present-day average annual surface temperature there is about -25 to -30 degrees Celsius

(Ettema et al., 2009).

2.36

p10, l17: "Little change in ice temperature over the time scale[. . . ]". I believe this to be incorrect. Bondzio et al. (2017) show that the ice stream can indeed warm by several degrees Celsius during the flow acceleration following the dis-integration of the ice tongue, which is significant for ice flow especially for the warm, soft ice near the terminus.

*Again, that study was published after our submittion, so we will refer to it and change our statement.*

2.37

p10, l25: "linear increase". Please elaborate exactly how the submarine melt rate increases linearly. From when to when? From which value to which value?

*As we state in the beginning of the paragraph, all parameters are changed linearly from 1850 to 2015 and all values are given in table 2. However, we will clarify this.*

2.38

p10, l27: "same gradual change in the SMB gradients from the LIA" is confusing to read as I they have been called "SMB profiles" in the caption of Fig. 2.

*Well, the gradient is the change along the x-axis, whereas the profile is a plot of the SMB values along the x-axis.*

2.39

p10, l31: "The sea ice buttressing can be assumed to be linearly dependent on the ocean and air temperatures[. . . ]". Please cite the observational study that motivates this choice. The decrease in sea-ice cover by a factor three is not obvious as of now.

*As far as we are aware, there is no study on the relation of sea ice buttressing and air and ocean temperatures. We write that it may have decrease by a factor three, which is used by Nick et al. (2013), but since this is an unknown parameter, we use a very large range for the sea ice buttressing factor, (from no change in sea ice buttressing to a reduction of 33%).*

2.40

p10, l33: "However, a temperature increase [. . . ]". This parameter choice multiplies the longitudinal strain rate by up to a factor of 3, which will largely increase calving (cf. Eqs. 3-6). It should be discussed as of how realistic the results using such high values are.

*In this context, we refer to air and ocean temperatures in the fjord, which both have increased by approximately 1.5C in the fjord. The calving however, is only related to melt water penetrating into the crevasses, not to temperatures.*

2.41

p11, Fig. 3: It is hard to read the exact values of the parameters used here in this 3D plot. Using a grid in the back planes may help. The exponent in the unit of the submarine melt rate is off. However, instead of this figure, I recommend using a table which lists all parameters and their values, and which

names the experiments.

*We can include all values in a table and add a grid to the figure. The exponent in the unit is due to a matlab bug and will be fixed.*

2.42

p11, l1-4: The description of observations belongs into Section 2. Please include citations. Also, it is usually better to describe temperature changes in absolute numbers, as a "50%" increase in temperature is ambiguous for the various temperature units in use.

*We will be more clear on the separation of observations and values used in our study. However, absolute temperatures are not used in our model, only values that are linked to temperature in a poorly known relation (crevasse-water-depth, submarine melt). Therefore, we are would like to refrain from stating absolute temperatures.*

2.43

p11: The physical motivation of the parameter choices for submarine melt and crevasse water depth is poor. It would make much more sense to me if the author would simply say: "In order to perform a sensitivity analysis to different parameters, we vary parameter X from . . . to . . . ". A sensitivity analysis should be in the center of every modeling study in order to gauge the robustness of the model results w.r.t. its input parameters (cf. also Enderlin et al., 2013).

*Okay, good suggestion, we will do this in the revised manuscript.*

2.44

p11, l7-11: "It is thereby tuned [. . . ]" From my understanding, figure 3 does not show how the parameters have been tuned to each other in order to achieve the observed retreat. Neither does it show their interdependence, as it only shows the parameter space used in this sensitivity study. Please clarify this statement.

*Figure 3 shows both the parameter space, but also that the parameters are dependent on each other, so that e.g. the needed crevasse-water-depth is determined by the values used for submarine melt rate and sea ice buttressing in order to achieve the observed retreat. We will clarify this.*

2.45

p12, l1: "well-known". This is an overstatement given the handful of calving front positions available from Bauer (1968) until the 1960s.

*Okay, we will reformulate this.*

2.46

p12, l1-6: Please rephrase, the message of how the forcing perturbations are constrained is unclear. Furthermore, it is unclear how you obtain the calving front positions w.r.t. your flowline. Do you mean you take them as the intersection of the flow line used in the model with the observed calving front positions?

Moreover, "latitudinal position" is an unsuitable coordinate, as JI's flow is not straight in the lowermost $\sim 60$ km.

*As we apply a linear increase in forcing parameters, we only tune the model to fit the observation during the LIA and in present-day and let the model glacier evolve freely in-between, only forced by the linear increase in forcing parameters. The observations for the time-period in between are not used as a model input, but for comparison with the model output.*

*We will explain this more detailed in the revised manuscript.*

*Will also change latitudinal position -> cross-trough position.*

2.47

p12, l22: "most reasonable". Please clarify.

*Okay: the forcing perturbations that are within a physical range.*

2.48

p12, l28 to p13, l2: "The position of Camp-2 [. . . ]". It is hard to understand from this sentence how the position of Camp 2 on a surface bump leads to the overestimation of the ice thickness. Please clarify. Please explain which surface has been smoothed, ideally already in the model setup section.

*We apologize for the misunderstanding. The surface has not been smoothed, but the model may overestimate surface bumps due to similar reasons as it overestimates the rapid retreat. Therefore the elevation of Camp-2 should rather be compared to an averaged height. We will rephrase this in the revised manuscript.*

2.49

p13, l7: "forced by [. . . ]". Please specify the parameters used for the forcing, as well as from when to when they have been increased linearly. In order to understand the results, the reader needs exact information on all forcings used in the model.

*The sentence is "forced by [...] SMB, submarine melt rate, crevasse water depth and reduction in sea ice buttressing". These are the parameters used for the forcing. The forcing is described in Section 4.2. In the results, where we explain that we only apply linear forcings starting in 1850 to 2012, with the parameter combinations given in Figure 3.*

2.50

p14, figure 5c. Please clarify, are these annual average or maximum velocities?

Please include the description of what is the grounding line flux (the circles?).

Please add what is the color-coding of the circles. Please remove one of the y-axes to the right of plot c). Please use y-axis limits so that the circles are contained within the frame. Which locations from Joughin et al. (2014) are used for the velocities? Could you show them in a map?

*Okay, we will improve Figure 5c and add a more detailed description of the observational data. The positions of the velocities are given in Joughin et al. (2014)—please see figure 1 and 2 therein—and we use yearly averaged values.*

2.51

p16, l8: "Our results highlight the importance [. . . ]". This is no novel result, compare e.g. Morlighem et al. (2016); Enderlin et al. (2013), who should be cited here.

*Okay.*

2.52

p17, l6: "a highly complex and non-linear response [. . . ]". This response is linked to the variations in glacier width and depth, and therefore scale with their respective complexity, isn't it? I would formulate it like that.

*Yes, thanks for the suggestion.*

2.53

Section 6.2 lacks a clear structure. It starts out by suggesting to use the model and fjord geometry to infer moraine positions, but then turns and wants to use moraine positions to understand the non-linear response of marine ice sheet margins. Both arguments remain unfinished, so please specify how you want to achieve either of your goals.

*We will rephrase this section. However, as we suggest that moraine positions are linked to grounding line stabilization, the argument can be turned in both ways:*
*The knowledge of fjord geometry or length of stabilization from model studies can give indications on moraine positions; conversely, big moraine systems indicate positions of grounding line stabilization.*

2.54

Moreover, while the geometric constrictions determine where the calving front stabilizes, there is little information on where the grounding line stabilizes, as this depends to a large degree on e.g. the ice stream's mass balance (i.e. lateral influx), the bedrock topography, and the treatment of the grounding line in the model (which remains unclear). Furthermore, as you state in your introduction, stable grounding lines on retrograde slopes are possible due to lateral stabilization (Gudmundsson et al., 2012). Last but not least, in order to answer whether moraines will form you need to know which process is dominant at a grounding line: substrate erosion or deposition. How can you tell that from your model? Given all the uncertainties in model input data and model parametriza-tions, please discuss how confident you can be in locations of grounding line stabilization obtained from your model, and hence moraine formation.

*The finding of our study that the knowledge on fjord geometry can help predict the position of past moraine positions in fjord systems is clearly novel. Therefore we write that "this hypothesis remains to be tested with a proper model of sediment dynamics and constrained by a number of well-studied, diverse glaciological and climatic environments." However, our model uses a detailed treatment of the grounding line (see comment 1.2) with a high resolution of less than 300 m and the ability to provide stable positions on both bedrock bumps and narrow sections, as shown in previous studies (Enderlin et al., 2013, Gudmundsson et al., 2012).*
*We will discuss the confidence of our hypothesis with our model and provide suggestions for further investigation in the revised manuscript.*

2.55

p18, l8: "can be predicted from geometric information[. . . ]". You mentioned further above (p17, l13)

that you cannot tell whether fjord width or depth is the dominant control on glacier retreat. Therefore, please specify which geometric information should be used to predict moraine formation, and how you can conclude that.

*We suggest that the width and depth are both important and the hypothesis needs to be tested further.*

2.56

p19, l14: "sea ice buttressing". How can an increase in sea ice buttressing trigger retreat?

*A reduction in sea ice buttressing can trigger retreat, we will change this in the revised manuscript.*

2.57

p19, l21-24: The message of these sentences is not clear to me. Please rephrase.

*Okay we will rephrase the explanation on an exaggerated crevasse water depth to account for the lack of submarine melt at the vertical calving front.*

2.58

p19, Sect. 6.3. The discussion of which model parameter the glacier model is most sensitive to should happen in section 6.1. Instead, in section 6.3 you should discuss the limitations to the conclusions you are drawing, see comments above.

*Yes thanks. We will move this part of the discussion.*

2.59

p19, l27: "SMB curve is lowered by 50 % [. . . ]". Lowering the SMB curve by 50% yields -4.5, not -6 m.w.e. yr$^{-1}$ at the terminus. Do you mean multiplying it by a factor two?

*No, the combination of doubling the frontal gradient (S_a1 in Equ.7) and lowering of the whole curve (a_0 in Equ.7) by 50% gives a SMB of -6 at the front.*

2.60

p20, l6: "For further studies [. . . ]". The reconstruction of the retreat has been done already by Bondzio et al. (2017), who should be cited here.

*Okay, we will include the citation.*

2.61

p20, Sect. 6.4 summarizes model results instead of discussing them. The results should be moved to the results section, and discussed here.

*As we do not intend to project the future of JI, we did not want to include this section in the results. We will, however, reformulate this section as implications for the future and for further projective modelling studies.*

**3 Minor Corrections**

*Thanks for the correction of minor mistakes. We will correct all of them once we got the second review and revise the manuscript.*

1. p2, l18: remains challenging

2. p2, l18: "constrains" feels wrong choice here. Perhaps better inhibits?

3. p2, l19: Here, we [. . . ]

4. p2, l21: [. . . ] provides the context for [. . . ]

5. p2, l27: "largest sea level contribution". Better: largest contribution to sea level rise.

6. p3, l5: "very limited": A matter of taste, but the emphasis caused by "very" can be omitted here.

7. p3, l20: "Section 6," An excess comma.

8. p3, l29: "is drained". A matter of taste again, but using active form is more engaging for the reader.

9. p4, l13: "still ongoing retreat". Perhaps better: "which is ongoing as of today".

10. p5, l11: "is contemporary to". Better perhaps: "coincides with"?

11. p7, l4: "Ice thickness changes with time are calculated [. . . ]". Please rephrase, "changes" reads as a verb, but is a noun, which is confusing to read.

12. p7, l7: "The mass balance B". B misses a dot.

13. p7, l24: "penetrates". Use plural here, as both surface crevasse depth and basal crevasse depth penetrate the glacier thickness.

14. p8, eq. 6: formatting: the dot over xx spreads over to the subscript characters.

15. p9, l6: "relatively" can be dropped.

16. p12, l16: "straightened" is repeated three times here, please rephrase.

17. p12, l19: "geometry". Perhaps better: model setup?

18. p12, l26: "(Gudmundsson, 2003)". As you are presenting your results here, the citation can be dropped in my opinion.

19. p13, l5: "surface pumps", typo, do you mean surface bumps?

20. p13, l11: "the glacier terminus changes". Please clarify what characteristic of the terminus changes. I assume it's the configuration?

21. p17, l18: "In addition [. . . ]" This starts a new topic, and deserves a new paragraph.

22. p19, l9: "among others those presented in Fig. 3". Please name them here for completeness. Also,

discuss here earlier studies that found a similar result (e.g. Enderlin et al., 2013).

23. p19, l19: "order of magnitude". This formulation is usually used with respect to magnitude, and is confusing in this sentence, as it misleads the reader to believe that the melting rate has to be multiplied by a factor 10 or so in order to achieve what a few percent in the crevasse water depth. I believe that you want to say is that the submarine melt rate has to be changed by tens of percent to achieve an effect that is reached by changing
the crevasse water depth by only a few percent, right? Please rephrase accordingly.

24. p21, l2: "Straightening". Please add: the bed.

**References**

*Boghosian, A., Tinto, K., Cochran, J. R., Porter, D., Elieff, S., Burton, B. L., and Bell, R. E.: Resolving bathymetry from airborne gravity along Greenland fjords, J. Geophys. Res-Sol. Ea., 119, 2015.*

*Bondzio, J. H., Morlighem, M., Seroussi, H., Kleiner, T., Rückamp, M., Mouginot, J., Moon, T., Larour, E. Y., and Humbert A.: The mechanisms behind Jakobshavn Isbræ's acceleration and mass loss: A 3-D thermomechanical model study. Geophys. Res. Lett., 44(12):6252–6260, 2017.*

*Bondzio, J. H., Seroussi, H., Morlighem, M., Kleiner, T., Rückamp, M., Humbert, A. and Larour, E.: Modelling calving front dynamics using a level-set method: application to Jakobshavn Isbræ, West Greenland. The Cryosphere, 10(2):497–510. 2016.*

*Cook, A. J., Holland, P. R., Meredith, M. P., Murray, T., Luckman, A., Vaughan, D. G.: Ocean forcing of glacier retreat in the western Antarctic Peninsula. Science, 283-286, 2016.*

*Cornford, S. L., Martin, D. F., Payne, A. J., Ng, E. G., Le Brocq, A. M., Gladstone, R. M., Edwards, T. L., Shannon, S. R., Agosta, C., van den Broeke, M. R., Hellmer, H. H., Krinner, G., Ligtenberg, S. R. M., Timmermann, R., and Vaughan, D. G.: Century-scale simulations of the response of the West Antarctic Ice Sheet to a warming climate, The Cryosphere, 9, 1579-1600, 2015.*

*Enderlin, E. M., Howat, I. M., and Vieli, A.: High sensitivity of tidewater outlet glacier dynamics to shape, Cryosphere, 7, 1007–1015, 2013.*

*Gillet-Chaulet, F., Gagliardini, O., Seddik, H., Nodet, M., Durand, G., Ritz, C., Zwinger, T., Greve, R., and Vaughan, D. G.: Greenland ice sheet contribution to sea-level rise from a new-generation ice-sheet model, The Cryosphere, 6, 1561-1576, 2012.*

*Gudmundsson, G. H., Krug, J., Durand, G., Favier, L., and Gagliardini, O.: The stability of grounding lines on retrograde slopes, Cryosphere, 2012.*

*Holland, D. M., Thomas, R. H., de Young, B., Ribergaard, M. H., and Lyberth, B.: Acceleration of Jakobshavn Isbræ triggered by warm subsurface ocean waters, Nat. Geosci., 1, 659–664, 2008.*

*Jamieson, S. S., Vieli, A., Livingstone, S. J., Ó Cofaigh, C., Stokes, C., Hillenbrand, C.-D., and Dowdeswell, J. a.: Ice-stream stability on a reverse bed slope, Nat. Geosci., 5, 799–802, 2012.*

*Joughin, I., Smith, B. E., Howat, I. M., Floricioiu, D., Alley, R. B., Truffer, M., and Fahnestock, M.: Seasonal to decadal scale variations in the surface velocity of Jakobshavn Isbrae, Greenland:*

*Observation and model-based analysis, J. Geophys. Res-Earth, 117, 1–20, 2012.*

*Joughin, I., Smith, B. E., Shean, D. E., and Floricioiu, D.: Brief Communication: Further summer speedup of Jakobshavn Isbræ, Cryosphere, 2014.*

*Khan, S. A., Aschwanden, A., Bjørk, A. A., Wahr, J., Kjeldsen, K. K., and Kjær, K. H.: Greenland ice sheet mass balance: a review, Rep. 2015.*

*Lea, J. M., Mair, D. W. F., Nick, F. M., Rea, B. R., Van As, D., Morlighem, M., Nienow, P. W., and Weidick, A.: Fluctuations of a Greenlandic tidewater glacier driven by changes in atmospheric forcing: Observations and modelling of Kangiata Nunaata Sermia, 1859-present, Cryosphere, 8, 2031–2045, 2014.*

*Lüthi, M., Funk, M., Iken, A., Gogineni, S., and Truffer, M.: Mechanisms of fast flow in Jakobshavn Isbrae, West Greenland: Part III. Measurements of ice deformation, temperature and cross-borehole conductivityin boreholes to the bedrock, J. Glaciol., 48, 369–385, 2002.*

*Mengel, M., Levermann, A., Frieler, K., Robinson, A., Marzeion, B. and Winkelmann, R.: Future sea level rise constrained by observations and long-term commitment. PNAS 113, 2016.*

*Morlighem, M., Bondzio, J., Seroussi, H., Rignot, E., Larour, E., Humbert, A. and Rebuffi, S.-A.: Modeling of Store Gletscher's calving dynamics, West Greenland, in response to ocean thermal forcing. Geophys. Res. Lett., 43(6):2659–2666, 2016.*

*Morlighem, M., Rignot, E., Mouginot, J., Seroussi, H., and Larour, E.: Deeply incised submarine glacial valleys beneath the Greenland ice, 2014.*

*Muresan, I. S., Khan, S. A., Aschwanden, A., Khroulev, C., Van Dam, T., Bamber, J., van den Broeke, M. R., Wouters, B., Kuipers Munneke, P., and Kjær, K. H.: Modelled glacier dynamics over the last quarter of a century at Jakobshavn Isbræ, The Cryosphere, 10, 597-611, 2016.*

*Nick, F. M., Vieli, A., Howat, I. M., and Joughin, I.: Large-scale changes in Greenland outlet glacier dynamics triggered at the terminus., Nat. Geosci., 2, 110–114, 2009.*

*Nick, F. M., Van Der Veen, C. J., Vieli, A., and Benn, D. I.: A physically based calving model applied to marine outlet glaciers and implications for the glacier dynamics, J. Glaciol., 56, 781–794, 2010.*

*Nick, F. M., Vieli, A., Andersen, M. L., Joughin, I., Payne, A., Edwards, T. L., Pattyn, F., and van de Wal, R. S. W.: Future sea-level rise from Greenland's main outlet glaciers in a warming climate., Nature, 497, 235–8, 2013.*

*Rignot, E. and Mouginot, J.: Ice flow in Greenland for the International Polar Year 2008-2009, Geophys. Res. Lett., 39, 1–7, 2012.*

*Shapero, D. R., Joughin, I. R., Poinar, K., Morlighem, M., and Gillet-Chaulet, F.: Basal resistance for three of the largest Greenland outlet glaciers, J. Geophys. Res-Earth, 121, 168–180, 2016.*

*Straneo, F. and Heimbach, P.: North Atlantic warming and the retreat of Greenland's outlet glaciers., Nature, 504, 36–43, 2013.*

*Thomas, R. H., Abdalati, W., Frederick, E., Krabill, W. B., Manizade, S., and Steffen, K.: Investigation of surface melting and dynamic thinning on Jakobshavn Isbræ, Greenland, J. Glaciol., 49, 231–239, 2003.*

van der Veen, C. J.: Tidewater calving, J. Glaciol., 42, 375–385, 1996.

van der Veen, C. J., Plummer, J. C., and Stearns, L. a.: Controls on the recent speed-up of Jakobshavn Isbræ, West Greenland, J. Glaciol., 57, 2011.

Vieli, A. and Payne, A. J.: Assessing the ability of numerical ice sheet models to simulate grounding line migration, J. Geophys. Res-Earth, 110, 1–18, 2005.

---

## Referee Comment (RC2) · M.P. Lüthi (Referee) · 15 Jan 2018

**Review: Steiger et al.; tc-2017-151**

Dear colleagues,

This is not a paper that is easy to judge. It presents a nice modeling study of the long-term retreat of one of Greenland's major glaciers with some nice figures. However, the results and conclusions are less convincing (except for the commonplace "geometry is important") than expected.

In my opinion the paper could be brought in a form that is interesting for the reader if the shortcomings of the model were worked out. As detailed below, the model formulation is too simple for the task at hand, and some of the parametrizations seem to fail. Or maybe only the forcing should be more realistic. Describing what goes wrong, and why, could provide important hints of the required model physics or parametrization.

Sincerely, Martin Lüthi

**General comments**

It is not clear what the authors want to achieve in this paper. The setup and the introductory sections target Jakobshavn IsbræThe results, however, do not match the measured evolution of this glacier, despite the arbitrary tuning of many model parameters to somehow achieve an agreement. The reasons for this mismatch are not investigated in the Discussion, but general observations are presented, that are not novel, and are also not clearly worked out. It is not clear by how much this study advances the topic since the many modeling papers of tidewater glaciers published during the last three decades, and notably those of F. Nick on this and other glaciers.

Many details on Jakobshavn Isbræ are given in the text. But then this complex glacier system is modeled with a code that lacks almost all features that were discussed in the sections before. It might well be that the general behaviour of tidewater glaciers can be captured by simple models (this is even true for much simpler models than Equation (2)), but real Jakobshavn Isbræ behaves differently than almost all assumptions implicitly stated in Section 3.1.

The presented model contains many tuning parameters with values that seem to be chosen *ad hoc*. This is not bad in general, but the predictive power of such a model is severely reduce since most physics is missing (ice flow, stress transfer, basal stress coupling, calving rates etc.) and just parametrized. I'm not generally opposed using simple models, but a very good rationale should be given (which is completely lacking in the introductory sections), and the approximations and parametrizations should be clearly stated.

Following the arguments for the parametrizations in Section 4.1 which all seem quite arbitrary, one wonders why such a complicated model has been used. Would a simpler model with less tuning parameters also do the job? Why care about ice temperature if viscosity is altered *ad hoc* with enhancement factors, and why care about water in crevasses if calving is somehow parametrized.

Section 4.2 continues arguing about parameters that are undetermined, and some *ad hoc* choices are made. Why care what these model parameters mean in real life? It might be worth a section in the Discussion, but most of Section 4.2 seems unnecessary and confusing. Nothing is known anyway, so why argue? Just clearly state what forcings are used, i.e. with explicit formulas that everyone can understand and repeat.

Also, I was missing the rationale for a linear forcing. While not a lot is known, at least for

temperature we have some ideas of the timing, and ocean temperatures increased almost step-wise around 1997. So it is likely that a more realistic forcing would provide more realistic results.

The Discussion seems to distract from the fact that the model (or the forcing) cannot be used for Jakobshavn (maybe because it is too complex), and looks at details of calving models (role of bed topography, glacier width, moraines) that have been treated in many papers.

Some newer literature (e.g. Felikson et al., 2017) should be included and discussed.

**Specific comments**

2/2 A 2012 paper seems outdated in this context.

2/4 specify: *surface runoff*

2/5 this is not as simple as said here. *might cause crevasses to penetrate deeper.* Whether this promotes calving (once the crevasses have been advected to the terminus) is also not so simple, since maybe long-during hydrofracturing actually drains crevasses, if links to the subglacial drainage system have been opened. Only if water supply starts close to the terminus, the process is very likely to enhance calving.

2/7 also Motyka et al. (2011)

2/13 not sure what "consistent" means here. Acceleration is not coupled to warming (there are indirect effects which can cause acceleration and/or deceleration).

2/30 "Destabilization": I would not call this destabilization, since the glacier is still stable, but retreating rapidly. Maybe in a dynamical systems representation, this you could discuss this in terms of stability, but this context is missing here. So better use *The rapid retreat…*

3/11 Carbonnell and Bauer (1968) have also flow velocity

3/25 or even century, as Rink, Wegener, Mercanton etc have measured its speed since 1875.

3/28 "see Fig": leave away "see".

3/30 a 2004 paper for discharge seems outdated in the context. The same formulation of page 6/28 should be adopted, but the same information is given twice.

3/31 "narrow" for 5 km wide?

3/32 cite Clarke and Echelmeyer (1996) here, who actually measured the trough. Morlighem's interpolations, while important, are sometimes very much off from measurements.

Fig1 it would be helpful to show the outlines of the fast-flowing ice stream for the reader unfamiliar with Jakobshavn.

4/9 It seems important tio mention that Jakobshavn had a long, floating terminus (e.g. Lingle et al. (1981); Motyka et al. (2011)) which rapidly disintegrated.

Fig2 indicate the 0 mb line, and also the ELA.

5/2 "has been reconstructed"

5/8 Fig 2 does not show changes, but average annual (?) values.

5/10 This number is pretty useless here, rather say by how much the local mass balance has changed at Jakobshavn.

5/11 "coincident"

6/7 "summer advances"? In Section 3.2, Casotto et al write about 5 km advance in winter

6/8 "shorter period"? I would think this is "longer" here.

6/19 The cited papers used the "old" Paterson values for $A$ in the flow law and enhanced deformation for Wisconsin ice. With that there is no need to invoke basal motion to account for the high observed velocities, depending on the assumed temperature profile. Lüthi et al. (2003) (Fig. 5) and Truffer and Echelmeyer (2003) showed that changing the basal resistance has a minor influence on basal stress field.

6/23 This was the whole point of Lüthi et al. (2003). But we and also Truffer and Echelmeyer (2003) independently obtained about 50% of driving stress, with two pretty good FE codes.

7/14 Standard use is $h$ or $H$ for thickness and $s$ for surface elevation.

Tab1 What is $A$? Which parametrization is assumed, traditional Patterson (1984), Cuffey and Patterson (2010), or anything else? Units for $A$ are wrong in any case (exponent should be $-3$).

8/4 two- and three-letter variable names are plain confusing. Better use $d_{cw}$, or similar

8/5 "fraction" *Rightarrow* "fracture"?

Eq8 These are very strong assumptions that need better motivation. They certainly are not proportional to ice stream speed, but rather to the elevation gradients between stream and sides.

9/18 Is Camp-2 identical with the SUSIE Air Greenland landing site, in vicinity of which also GPS station KAGA is located. If so, please rename accordingly, also in [12/23].

8/18 Why would one assume a steady state around 1850? In the introduction the whole history since the ice age was laid out, and I'm convinced that there was never a steady state.

10/15 What is averaged, the temperature, or the rate factor which varies exponentially with temperature? It seems that the temperature is averaged which seems not very relevant for ice deformation studies. Also note that most deformation happens in the bottom 20% (or so) of the ice column, mainly due to high shear stress and high temperature (up to temperate).

And maybe a more important question: is the averaging of $A$ done for the horizontal stress transfer (which is dominated by the very cold ice) or the vertical shearing (which is dominated by the bottom warm temperatures)?

8/25 Obviously, anything enhancing crevasses will reduce the glacier length. The statement, however, is unclear. Are all quantities changed at once? Why is the result stated before the experiment is described?

10/30 Are these three parameters independent of each other in the model description? Since I did not double-check, it would be nice if the authors would provide this information, and also show what individual changes in these parameters do to the glacier.

11/1 What does this mean: "temperature has doubled"? From $272\,\mathrm{K}$ to $544\,\mathrm{K}$. Please give absolute values, temperature percentage is meaningless in this context.

11/5 Why care about water in crevasses? Nothing is known (except: there is no water in crevasses for 3/4 of the year), so you are free to force the model with whatever works.

13/1 Trim lines are usually lower than the center line height of a glacier. SUSIE/KAGA is also not on the central branch, but probably mostly affected by the (former, until 2010) North branch, which is completely ignored in the model.

13/7 Are all parameters varied simultaneously? It would by very helpful to just give the formulas for these changes with time.

Fig6 Very difficult to see the different colors, and match them back to Fig. 3.

15/1 This is a very generic property of a nonlinear system. Tidewater glaciers with over-deepened beds are good examples thereof, and this remark, re-iterated several times, is not at all novel or surprising.

15/13 Now, this stability investigation would be interesting if done correctly (in the sense of dynamical system analysis).

Fig7 So we see that the glacier during its retreat rests on narrow hills, and rapidly transits through wide depressions. In my opinion no surprises here. Analyzing your system equations (1, 2), you would find exactly that.

16/9 or even: stochastic or no forcing at all. This is well known, with a special variant termed by Post "the tidewater glacier cycle".

21/5 A very unclear statement, since moraines *are* part of the (basal) geometry.

**References**

Carbonnell, M. and Bauer, A. (1968). Exploitation des couvertures photographiques aériennes répétées du front des glaciers vêlants dans Disko Bugt et Umanak Fjord, Juin-Juillet 1964. Technical Report 3, Expédition glaciologique internationale au Groenland (EGIG). Tirage à part des Meddelelser om Grønland, Bd. 173, Nr. 5.

Clarke, T. S. and Echelmeyer, K. (1996). Seismic-reflection evidence for a deep subglacial trough beneath Jakobshavns Isbrae, West Greenland. *Journal of Glaciology*, 43(141):219–232.

Felikson, D., Bartholomaus, T. C., Catania, G. A., Korsgaard, N. J., Kjær, K. H., Morlighem, M., Noël, B., van den Broeke, M., Stearns, L. A., Shroyer, E. L., Sutherland, D. A., and Nash, J. D. (2017). Inland thinning on the greenland ice sheet controlled by outlet glacier geometry. *Nature Geoscience*, 10:366–369.

Lingle, C. S., Hughes, T. J., and Kollmeyer, R. C. (1981). Tidal flexure of Jakobshavns Glacier, West Greenland. *Journal of Geophysical Research*, 86(B5):3960–3968.

Lüthi, M. P., Funk, M., and Iken, A. (2003). Indication of active overthrust faulting along the Holocene-Wisconsin transition in the marginal zone of Jakobshavn Isbræ. *Journal of Geophysical Research*, 108(B11).

Motyka, R. J., Fahnestock, M., Truffer, M., Mortensen, J., and Rysgaard, S. (2011). Submarine melting of the 1985 Jakobshavn Isbræ floating tongue and the triggering of the current retreat. *Journal of Geophysical Research*, 116(F01007).

Truffer, M. and Echelmeyer, K. (2003). Of Isbræ and Ice Streams. *Annals of Glaciology*, 36:66–72.

---

## Author Comment (AC2) · 13 Feb 2018

Reply to comments from January 15, 2018 by M.P. Lüthi
on N. Steiger et al., "Non-linear retreat of Jakobshavn Isbræ. . . "

https://doi.org/10.5194/tc-2017-151-RC2

February 14, 2018

Review: Steiger et al.; tc-2017-151

Dear colleagues,
This is not a paper that is easy to judge. It presents a nice modeling study of the long-term retreat of one of Greenland's major glaciers with some nice figures. However, the results and conclusions are less convincing (except for the commonplace "geometry is important") than expected.

*We would like to thank you for your input and would like to reply to your arguments in the following. In this study, we present three very important points that are crucial to be considered in any reconstruction and projection study and that we will make more clear in the revised version:*

- *Geometric pinning points—defined by the trough bed and width—determine the position of grounding line stabilization during retreat and advance of the glacier, whereas climate forcing determines the timing of retreat from a stable position.*
- *A delayed rapid glacier retreat can result from temperature changes taking place decades ago due to the long internal response time and the stabilization on pinning points.*
- *We present the novel idea that narrow sections in the glacial trough can give the position of moraines, which providing a guide for the work of geomorphologists.*

*To clarify the novelty of the findings we would like to point to our response to the first review:*

*There are indeed other studies showing the importance of bedrock geometry (e.g. Schoof 2006; Enderlin et al 2012; Jamieson et al. 2012; 2013; Morlighem et al. 2016). However, we wouldn't say that its is commonplace and most of these studies focus on synthetic glacier geometries, without a model validation using realistic changes to the external forcing and the width- depth size ratio. Also, in the community there is a strong emphasis on the role of ice-ocean interactions as a key control on the retreat of marine terminating glaciers such as JI (e.g. Holland et al., 2008; Joughin et al., 2012; Straneo and Heimbach, 2013; Cook et al., 2016; Mengel et al., 2016). In particular, it is often implied that changes in climate, ocean circulation and temperature are the main controls on glacier retreat.This is only partly true, as once a retreat at a marine margin is triggered, the influence of bedrock geometry and fjord width dominate the transient response of the glacier, as is shown in our study as well as others (e.g. Jamieson et al. 2012; 2013; Morlighem et al. 2016).*

*In addition, most studies only consider the recent retreat of glaciers such as JI and not the long term response where fjord geometry dominates the transient response (e.g. Nick et al., 2013; Muresan et al., 2015; Bondzio et al., 2017). In this study we consider the period starting at the Little Ice Age which clearly is novel for JI and gives a significant increase in the data available to study the long term response of marine terminating glaciers such as JI. We would claim that only using the recent observed retreat might be misleading and is not adequate for understanding the long term retreat of marine terminating glaciers in Greenland, in particular because fjord geometry is such an*

*important factor on longer timescales. In addition, as pointed out in the paper, past changes in climate could be more important than contemporary changes in triggering the recent observed retreat given the long timescales involved. Similarly, current changes in climate may trigger a delayed rapid retreat in the future for glaciers presumed stable.*

*However, perhaps the most novel finding of our study is that an understanding of the relative impact of fjord geometry can help predict the position of past moraine positions in fjord systems such as Jakobshavn. This information will be particularly valuable in assessing the geometry and glacial history of similar fjord systems such as those in Greenland, Norway, Patagonia, and Alaska.*

*We have revised the paper accordingly to make these novel points clear to the reader.*

In my opinion the paper could be brought in a form that is interesting for the reader if the shortcomings of the model were worked out. As detailed below, the model formulation is too simple for the task at hand, and some of the parametrizations seem to fail. Or maybe only the forcing should be more realistic. Describing what goes wrong, and why, could provide important hints of the required model physics or parametrization.
Sincerely, Martin Lüthi

*Thanks for the suggestions of how to improve the paper. In the revised manuscript we now discuss the shortcomings of the model in greater detail than was done in the original Section 6.3. In addition to the limitations of a simple model we also note that the 1D flowline model has its advantages. In particular, it is extremely efficient, making it possible to run a very large number of ensemble experiments exploring the impact of different forcing factors and choices of parameter values (as an example, more than 2000 simulations each 250 model years were completed during this study, only a subset of the most relevant ones are shown in the paper). In addition to this, the model includes an explicit, physical treatment of calving and frontal dynamics which allows for the study of transient frontal retreat, given changes in the climate forcing (submarine melt, sea ice buttressing, surface melt and crevasse water depth). In many studies, including those with 3D dynamical flow models, the frontal position through time is not explicitly resolved, and is rather fixed in time or forced to a particular position manually (e.g. Gillet-Chaulet et al., 2012; Cornford et al., 2015; Bondzio et al., 2017). Also, the parametrizations are physically based and link processes that are still not completely understood to air and ocean temperatures. However, as the past changes in climate are not accurately known we choose a simple linear increase to be able to better isolate the glacier responses to geometric effects.*

**General comments**
It is not clear what the authors want to achieve in this paper. The setup and the introductory sections target Jakobshavn Isbræ. The results, however, do not match the measured evolution of this glacier, despite the arbitrary tuning of many model parameters to somehow achieve an agreement.

*In the revised manuscript, we removed the background section on JI to avoid confusion on why we apply the model to JI. The aim of this study is to study the external, glaciological and geometric controls on JI in response to a linear forcing on long time scales. Rather than a simulation tuned perfectly to the observed retreat history of the past 150 years of JI, we perform a sensitivity study to investigate the importance of the fjord geometry of JI in governing the response of the glacier. However, the model parameters are not arbitrary, as the application of the flowline model to JI constrains the parameters so that the modelled velocities and frontal positions during the LIA and today correspond to observations from JI. As parameters such as crevasse water depth and sea ice*

*buttressing cannot easily be quantified, it is important to relate them to observed values such as the calving rate.*

The reasons for this mismatch are not investigated in the Discussion, but general observations are presented, that are not novel, and are also not clearly worked out. It is not clear by how much this study advances the topic since the many modeling papers of tidewater glaciers published during the last three decades, and notably those of F. Nick on this and other glaciers.

*Given the simplicity of the model and the linear climate forcing applied it is surprising that the flow line can reproduce the highly nonlinear retreat history of JI over the past 150 years. Still, as pointed out, there is a mismatch with observations as discussed in original Section 6.3. This discussion is now expanded giving more details as requested.*

*Note that it is not our intension to reproduce the retreat history in detail—for this a more complex model is required (e.g. Bondzio et al., 2017), as well as an accurate history of changes in climate (which is not available back to 1850). Instead, our intentions are to investigate the source of the highly non-linear response of the glacier despite a linear climate forcing. Note also that the original studies of F. Nick and others did not include the longer retreat history in their analysis as done in this paper, instead these earlier studies were focused on the more recent as well as future retreat of JI.*

Many details on Jakobshavn Isbræ are given in the text. But then this complex glacier system is modeled with a code that lacks almost all features that were discussed in the sections before. It might well be that the general behaviour of tidewater glaciers can be captured by simple models (this is even true for much simpler models than Equation (2)), but real Jakobshavn Isbræ behaves differently than almost all assumptions implicitly stated in Section 3.1.

*As noted above, the manuscript is now revised and the background on JI has been removed to avoid confusion. We are aware that JI is a complex glacier, which makes it even more surprising that the disintegration of its floating tongue can be modeled (although exaggerated) by linearly changing crevasse water depth, SMB, submarine melt rate and sea ice buttressing.*

The presented model contains many tuning parameters with values that seem to be chosen ad hoc. This is not bad in general, but the predictive power of such a model is severely reduce since most physics is missing (ice flow, stress transfer, basal stress coupling, calving rates etc.) and just parametrized. I'm not generally opposed using simple models, but a very good rationale should be given (which is completely lacking in the introductory sections), and the approximations and parametrizations should be clearly stated.

*Most models include tuning parameters and most of the parameters used in our model are linked to air or ocean temperatures, and serve as forcing parameters with magnitudes linked to observations  (see Sections 4.1 and 4.2). The parametrizations for the stress balance and calving are physically based. Note that there is still no clear consensus how to correctly implement calving, buttressing and submarine melt rate in models. We are confident that the parameter choices we made to account for the long term history of JI give a reasonably good representation of its behaviour. In the revised manuscript details of the tuning parameters are given in Section 2.*

Following the arguments for the parametrizations in Section 4.1 which all seem quite arbitrary, one wonders why such a complicated model has been used. Would a simpler model with less tuning parameters also do the job?

*For the purpose of studying the impact of fjord geometry, it is necessaire to include a free evolution of a floating tongue (Vieli et al., 2001), a physical calving law (Benn et al., 2007; Nick et al., 2010) and a robust treatment of the grounding line with a moving grid (Vieli and Payne, 2005). The used parameters allow the application of a climate forcing and the tuning of the model using observational data. The physically-based parametrizations and the corresponding parameters are described in Sections 2 and 3 in the revised manuscript.*

Why care about ice temperature if viscosity is altered ad hoc with enhancement factors, and why care about water in crevasses if calving is somehow parametrized.

*Ice temperature is not implicitly included in the model. Instead it is used to chose the right rate factor. The viscocity is not changed ad hoc, reather it is calculated via the strain rate and rate factor (Nick et al. 2010). Regarding calving, all models for marine-terminating glaciers require a parametrization of this process. The crevasse-depth criterion is one such parametrization which has been extensively tested by Nick et al., 2010. It calculates the opening of crevasses as a consequence of tensile stresses. The role of water in crevasses as a link to climate forcing is described in Section 2.2.*

Section 4.2 continues arguing about parameters that are undetermined, and some ad hoc choices are made. Why care what these model parameters mean in real life? It might be worth a section in the Discussion, but most of Section 4.2 seems unnecessary and confusing. Nothing is known anyway, so why argue? Just clearly state what forcings are used, i.e. with explicit formulas that everyone can understand and repeat.

*In our opinion, it is important to use reasonable values for the parameters, so that the retreat of the model glacier as far as possible reproduces the observe retreat history and velocities of JI. We have included a clear presentation of which parameters are used in Table 1 and Section 3 of the revised manuscript.*

Also, I was missing the rationale for a linear forcing. While not a lot is known, at least for temperature we have some ideas of the timing, and ocean temperatures increased almost step-wise around 1997. So it is likely that a more realistic forcing would provide more realistic results.

*Yes, we agree! A more realistic forcing would provide a more realistic result. But we are applying a linear forcing and a step forcing to show that rapid retreat does not need to be caused by suddenly higher temperatures, but instead can be caused solely by the geometry. When applying a more realistic forcing, it becomes difficult to argue wether the forcing or the geometry cause the non-linearity. Also— as you correctly pointed out before—the parameters are not directly linked to temperature, so that a more realistic forcing would just add more unknown complexity.*

The Discussion seems to distract from the fact that the model (or the forcing) cannot be used for Jakobshavn (maybe because it is too complex), and looks at details of calving models (role of bed topography, glacier width, moraines) that have been treated in many papers.

*In the revised manuscript we elaborate on the reasons for the deviation of the simulated retreat from the observations. However, as mentioned, it is not the aim of the study to reproduce 150 years of retreat history of JI, it is rather to use the long history of frontal positions to constrain the range of*

*parameters choices in the model. Also, although the importance of bed topography and glacier width is known in the community, many studies use short time periods to train their models (e.g. Nick et al., 2013; Muresan et al., 2015; Bondzio et al., 2017), disregarding the potential long-term glacier response to accumulated changes in climate. It is also true that many previous studies focus mostly on the ocean and atmosphere as drivers of rapid retreat of glaciers such as JI (e.g. Holland et al., 2008; Joughin et al., 2012; Straneo and Heimbach, 2013; Cook et al., 2016; Mengel et al., 2016).*

Some newer literature (e.g. Felikson et al., 2017) should be included and discussed.

*Thanks. We have also include other recent literature (such as Morlighem et al., 2016; Bondizio et al., 2017; Felikson et al., 2017) in the revised manuscript.*

**Specific comments**
2/2     A 2012 paper seems outdated in this context. *Thanks, this is updated.*

2/4     specify: surface runoff *OK, thanks*

2/5     this is not as simple as said here. might cause crevasses to penetrate deeper. Whether
this promotes calving (once the crevasses have been advected to the terminus) is also
not so simple, since maybe long-during hydrofracturing actually drains crevasses,
if links to the subglacial drainage system have been opened. Only if water supply
starts close to the terminus, the process is very likely to enhance calving.

*You are right that calving and hydrofracturing is a complex process, which is not fully understood. However, it is a key process which must be implemented in models of marine-terminating glaciers and ice sheets. Here, we adopt the the crevasse-water depth criterion (Nick et al., 2010) which is a physically based parameterization of calving. However, note that crevasse water depth is one of several parameters destabilizing the glacier front and increasing the calving rate. It is also true that it is mostly valied close to the terminus. This point has been adressed in the discussion of the revised manuscript.*

2/7     also Motyka et al. (2011) *thanks*

2/13    not sure what "consistent" means here. Acceleration is not coupled to warming
(there are indirect effects which can cause acceleration and/or deceleration). *Thanks, consistent -> correlated*

2/30    "Destabilization": I would not call this destabilization, since the glacier is still stable,
but retreating rapidly. Maybe in a dynamical systems representation, this you could
discuss this in terms of stability, but this context is missing here. So better use The
rapid retreat...                          *Okay, thanks*

3/11    Carbonnell and Bauer (1968) have also flow velocity   *The full paper was unfortunately not available to us, which is why we did not included the citation.*

3/25    or even century, as Rink, Wegener, Mercanton etc have measured its speed since 1875.

3/28    "see Fig": leave away "see". *Thanks*

3/30     a 2004 paper for discharge seems outdated in the context. The same formulation of page 6/28 should be adopted, but the same information is given twice.  *We will include more recent papers*

3/31     "narrow" for 5 km wide? *Relative to the depth it is narrow, but you are right we won't call it narrow*

3/32     cite Clarke and Echelmeyer (1996) here, who actually measured the trough. Morlighem's interpolations, while important, are sometimes very much off from measurements.
Fig1 it would be helpful to show the outlines of the fast-flowing ice stream for the reader unfamiliar with Jakobshavn.
*We will cite them, but we used Morlighem's data, since they are the best once that existed for the whole trough at the beginning of the construction of this study.*

4/9     It seems important tio mention that Jakobshavn had a long, floating terminus (e.g. Lingle et al. (1981); Motyka et al. (2011)) which rapidly disintegrated.
*This is  mentioned in 4/10 in the original manuscript.*
Fig2     indicate the 0 mb line, and also the ELA.
*Fig2 is removed in the revised paper.*

5/2     "has been reconstructed" *thanks*

5/8     Fig 2 does not show changes, but average annual (?) values.  *Yes, annual data averaged over a 10 year period. Fig 2 is removed.*

5/10 This number is pretty useless here, rather say by how much the local mass balance has changed at Jakobshavn.
*The value for the increase in surface runoff is used for the crevasse water depth. This is explained better in the revised Section 4.2*

5/11     "coincident" *thanks*

6/7     "summer advances"? In Section 3.2, Casotto et al write about 5 km advance in winter
*Sorry, that was a typo, thanks for correcting*

6/8     "shorter period"? I would think this is "longer" here. *That's right, thanks*

6/19     The cited papers used the "old" Paterson values for A in the flow law and enhanced deformation for Wisconsin ice. With that there is no need to invoke basal motion to account for the high observed velocities, depending on the assumed temperature profile. Lüthi et al. (2003) (Fig. 5) and Truffer and Echelmeyer (2003) showed that changing the basal resistance has a minor influence on basal stress field.

*I think there is no clear consensus on the influence of a change in basal resistance. Here, we would like to reflect on our response to treviewer 1:*
*Reviewing the literature, there is no clear consensus on the strength of the lateral and basal shear at JI, especially in a width-averaged setting. Some studies find that the base in the trough is very slippery, explaining the high velocities (Luthi et al., 2002; Shapero et al., 2016, Bondzio et al., 2017). However, these are observations at only one distance from the terminus (Luthi et al) and an inverse*

*model that does not include any weakening at the lateral shear margins through crevassing and meltwater penetration (Shapero et al), which is suggested as main reason for the high velocities elsewhere (Van der Veen, 2011). Joughin et al. (2012) only find a very weak bed in the trough using a stiff model, which is in disagreement with the observations, whereas they find a less pronounced difference between the basal resistance in the trough and the margins, when using a soft model. We agree on that models that are used to project the future of JI should be more complex and include a full representation of the lateral shear.*

6/23     This was the whole point of Lüthi et al. (2003). But we and also Truffer and Echelmeyer (2003) independently obtained about 50% of driving stress, with two pretty good FE codes.

*As shown in the comment above, there is no real consensus about the strength of the driving stress, but a short discussion is included in Section 5.3 of the revised manuscript.*

7/14     Standard use is h or H for thickness and s for surface elevation. *Okay, this is being changed.*
Tab1 What is A? Which parametrization is assumed, traditional Patterson (1984), Cuffey and Patterson (2010), or anything else? *Yes, as stated in the table caption.*
Units for A are wrong in any case (exponent should be −3). *okay, thanks*

8/4      two- and three-letter variable names are plain confusing. Better use d cw , or similar
*Okay, all parameters with several letters are renamed.*

8/5      "fraction" Rightarrow "fracture"? *Thanks.*

Eq8     These are very strong assumptions that need better motivation. They certainly are
not proportional to ice stream speed, but rather to the elevation gradients between
stream and sides.
*This is the best we can do in a one-dimensional model and it is based on mass conservation.The flux that enters the main trough from the sides has to exit through the main trough. It is reformulated in the revised version.*

9/18     Is Camp-2 identical with the SUSIE Air Greenland landing site, in vicinity of which
also GPS station KAGA is located. If so, please rename accordingly, also in [12/23].

*Thanks for this comment. The name of the positions is changed to KAGA in the revised manuscript.*

8/18 Why would one assume a steady state around 1850? In the introduction the whole
history since the ice age was laid out, and I'm convinced that there was never a
steady state.

*No you are right, this was an unclear formulation.But it was in a transition between advance and retreat, which makes LIA a natural starting point for the simulations.*

10/15   What is averaged, the temperature, or the rate factor which varies exponentially
with temperature? It seems that the temperature is averaged which seems not very
relevant for ice deformation studies. Also note that most deformation happens in
the bottom 20% (or so) of the ice column, mainly due to high shear stress and high
temperature (up to temperate).

*The temperature only goes into the model indirectly via the rate factor, which is kept constant. If temperature is constant, the rate factor is constant as well. Yes, most deformation happens at the bottom, but the model is depth averaged and we are only interested in the main ice flow along the flow line.*

And maybe a more important question: is the averaging of A done for the horizontal stress transfer (which is dominated by the very cold ice) or the vertical shearing (which is dominated by the bottom warm temperatures)?

*We will stress that a constant rate factor is used in depth and width, both for the calculation of the viscosity and the longitudinal stress gradient.*

8/25    Obviously, anything enhancing crevasses will reduce the glacier length. The statement, however, is unclear. Are all quantities changed at once? Why is the result stated before the experiment is described?

*Yes, the parameters are changed linearly and at the same time. It is not a results that the parameters force the model to retreat 43km, but a forcing constraint. Only the parameter combinations that achieve this retreat are used here. This has been elaborated in Section 3.2 of the revised manuscript.*

10/30   Are these three parameters independent of each other in the model description? Since I did not double-check, it would be nice if the authors would provide this information, and also show what individual changes in these parameters do to the glacier.

*The parameters themselves are independent, but they all cause a glacier retreat, so that the choice of each parameter depends on the choice of the other parameters to simulate the 43 km retreat since the LIA. We have included  this information in the revised manuscript (Section 3.2)*

11/1    What does this mean: "temperature has doubled"? From 272 K to 544 K. Please give absolute values, temperature percentage is meaningless in this context.

*Thanks, we have included absolute values. (From 1.5C to 3C)*

11/5    Why care about water in crevasses? Nothing is known (except: there is no water in crevasses for 3/4 of the year), so you are free to force the model with whatever works.

*You are right that nothing is known, but we know that surface runoff has increased by 63% and that calving rates have approximately doubled (Joughin 2004). The values for calving rates vary among different studies, but that's why we apply a large range.*

13/1    Trim lines are usually lower than the center line height of a glacier. SUSIE/KAGA is also not on the central branch, but probably mostly affected by the (former, until 2010) North branch, which is completely ignored in the model.

*Okay thanks, we didn't think about that. It may explain why our surface is larger at that position than the observed trimline.*

13/7    Are all parameters varied simultaneously? It would by very helpful to just give the
formulas for these changes with time.
Fig6 Very difficult to see the different colors, and match them back to Fig. 3.

*Yes, they are changed simultaneously. They are only changed linearly and the rate of change is given in Table 2 of the revised manuscript. Fig 3 is removed, and the colors are consequently arbitrary.*

15/1    This is a very generic property of a nonlinear system. Tidewater glaciers with over-
deepened beds are good examples thereof, and this remark, re-iterated several times,
is not at all novel or surprising.

*As mentioned earlier, we largely relate to the trough width and find that there is no clear consensus in the community on the link of the non-linear retreat to trough geometry. Many studies try to explain rapid retreat solely based on changes in temperatures.*

15/13   Now, this stability investigation would be interesting if done correctly (in the sense
of dynamical system analysis).
Fig7 So we see that the glacier during its retreat rests on narrow hills, and rapidly transits
through wide depressions. In my opinion no surprises here. Analyzing your system
equations (1, 2), you would find exactly that.

*We are not sure what you mean by dynamic system analysis. We use a novel way of quantifying grounding line stability and the figures show clearly the difference between the different geometries.*

16/9 or even: stochastic or no forcing at all. This is well known, with a special variant termed by Post
"the tidewater glacier cycle".

*Yes, also stochastic or no forcing at all. This is included in the revised manuscript. Although the non-linear retreat of marine-terminating glaciers is known, it is still an interesting exercise to pinpoint the origin of the non-linearity. Also, the tidewater glacier cycle rather describes the lengthening and shortening of the glacier tongue, whereas we refer to stabilization of the grounding line.*

21/5 A very unclear statement, since moraines are part of the (basal) geometry.

*True, but we say that they build  close to geometrical pinning points, and that the trough width can help finding positions of moraines. This has been revised in the new manuscript.*

---

## Author Comment (AC3) · 13 Feb 2018

The comment was uploaded in the form of a supplement:
https://www.the-cryosphere-discuss.net/tc-2017-151/tc-2017-151-AC3-supplement.pdf
* * *

---

## Referee Report (RR1)

**Comments on N. Steiger et al., "Non-linear retreat of Jakobshavn Isbræ…" https://doi.org/10.5194/tc-2017-151**

J. H. Bondzio

April 3, 2018

**1 General Comments**

**1.1 Summary**

N. Steiger et al. set up a 1D flowline glacier model of Jakobshavn Isbræ (JI) and perform a sensitivity analysis on various climatic and geometric model input parameters for the glacier's evolution from the Little Ice Age to today and into the future (until 2100). The authors conclude that the fjord and trough geometry are the main controls on the glacier's retreat history. They argue to be able to infer points of grounding line stabilization – and hence moraine formation – from the trough geometry using an ice flow model.

**1.2 Novelty**

The study has two main threads:

The first is the "non-linear" dynamic response of the glacier, controlled by the bed topography, once the calving front retreat has been triggered. The idea that bed topography controls the calving front retreat is not new (cf. e.g. Enderlin et al., 2013; Morlighem et al., 2016), and the study does not provide substantially more information beyond this statement. The thread in its current form should therefore be dropped. It would be worthwhile to quantify and analyze the degree of non-linearity of the glacier response, but this is difficult with the irregular real-world glacier geometry used here. An idea would be to use an artificial bed topography and to quantify exactly how the bed topography translates into a front retreat rate. When using this approach, it would need to be discussed how much of the response is due to model physics and how much due to model parametrizations.

The second thread is the argument to infer stable grounding line positions by combining an ice flow model with geomorphological information, i.o. to infer potential sites for moraine formation. This thread is novel (to my knowledge).

Two comments on this: First, I'd suggest to motivate more clearly why it is important to identify these sites for readers that are not familiar with the subject. Second, I have reservations about the practicability of the method, cf. section 1.3.

**1.3  Inferring Stable Grounding Line Positions Using an Ice Flow Model**

The second thread of the paper assumes that it is possible to determine the exact grounding line position over time from an ice sheet model only. I question this assumption for the following reasons:

- First, there is a large spread in the grounding line position across different ice flow models – and even for different mesh resolutions within the same model – for otherwise equal model setups (cf. e.g. Pattyn et al., 2013).

- Second, observational errors in bed topography and ocean melting rate near the grounding line are large. However, ice flow models are highly sensitive to small errors in exactly these model input parameters.

Hence, the likelihood of predicting the 'correct' grounding line position using only one model is small. The likelihood decreases further with the duration of the simulation, as errors add up and are amplified by dynamic englacial non-linear processes.

**1.4  Model Fitness**

I have concerns that the 1D flowline model used here is not able to accurately capture the "correct" stable grounding line positions, as important physical processes for grounding line stabilization such as lateral stabilization (cf. e.g. Gudmundsson et al., 2012) are missing or parameterized at best. The study would have to show that the setup presented here is able to match the grounding lines obtained with 2D or 3D ice flow models. If the authors choose to only present the idea of the second thread here, I'd suggest to discuss which models would be better suited to capture the grounding line in future work.

My current understanding on numerical modelling of JI and other isbræ-type outlet glaciers is that lateral physical effects (stress transfer, mass influx) have to be explicitly modeled due to their high importance for the glacier dynamics and their capacity for rapid, non-linear change themselves (Truffer and Echelmeyer, 2003; Joughin et al., 2012; Shapero et al., 2016; Bondzio et al., 2017). The model results obtained from using a 1D flowline model as used here have therefore to be interpreted with care, especially since some of the model parameters used here are unphysical (cf. specific comment 2.2).

**2   Specific Comments**

**2.1**

p2, l10:"Compared to previous studies [..]": A review of modelling studies and their findings that treat the same problem (JI) is lacking. A few studies that you might want to discuss are Truffer and Echelmeyer (2003); Vieli and Nick (2011); Joughin et al. (2012); Enderlin et al. (2013); Muresan et al. (2016); Shapero et al. (2016); Bondzio et al. (2017). In particular, the main differences to Enderlin et al. (2013) have to be pointed out.

**2.2**

p5, Table 1: The model uses unphysical model parameters. Crevasse water depths of up to 395 m are higher than observed, and submarine melting rates of only 175 m/a are lower than observed (Motyka et al., 2011). Moreover, these two model parameters tend to influence calving in the same way (higher respective values lead to higher calving rates). Why have they thus not been chosen in the range of observed values? Please motivate your model parameter choice.

**2.3**

p7, Eq.5: What are $Q_{JI,0}$ and $Q_{JI,t}$? Please discuss that this scaling of the lateral ice influx allows only for small perturbations in mass flux, as geometric changes will alter the lateral influx along the ice stream over time. If, for example, the ice flow velocity of JI doubles, your lateral influx will double as well, which is contrary to what happens when you model the lateral physics explicitly: then, a thinning ice stream thins the surroundings of an ice stream, which (initially) reduces the mass flux into the ice stream. Your parametrization of lateral influx therefore potentially "overfeeds" the ice stream in comparison. The motivation and discussion of this parametrization is important, as the lateral mass influx affects the grounding line position directly.

**2.4**

p9, l14,15: It is not clear to me why you use the "mean latitudinal position" of each calving front? Moreover, if you mean the latitudinal coordinate of each calving front position, the please explain how you deal with the fact that the glacier trough is bent: fronts along a North-South oriented section of the trough would then receive the same "latitudinal position".

**2.5**

p15, l20-22: The idea presented here is not new, cf. e.g. Vieli and Nick (2011).

**3 Minor Corrections**

1. p2, l32: "still": This is either a typo or it suggests that you do not agree with the hypothesis that the ocean has an influence on the glacier. Please clarify.

2. p5, table 1: The dot on $\epsilon_{xx}$ is misplaced.

3. p3, l2: "long timescale": Please be more specific. A centennial or decadal time scale?

4. p6, Eq. 3: The equation interrupts the text flow. Section 2.2 needs to be restructured for text flow.

5. p6, Eq. 4: Due to hydrostatic equilibrium, $D = \rho_i/\rho_w H$. Hence, Eq. 4 can be simplified to the form of Eq. 6 in Enderlin et al. (2013).

6. p6, l14: The model variable SMB, $a$, is usually put between two commata.

7. p7, l8: "The intention". This sentence is incomplete. The intention is to use a realistic geometry to do what exactly?

8. p7, l17: "bed topography profile": This is a repetition of p7, l9.

9. p8, l1: Bondzio et al. (2017) showed that the study attributes the glacier's high flow velocities to the interplay of both the slippery bed and the dynamically weakening shear margins, not just a slippery bed.

10. p8, l2: There is a question mark at the location of the citation in the text.

11. p8, l26: "outside JI". Please clarify.

12. p9, l4: The sea-ice buttressing in the model is an enhancement factor for the calving rate (Eq. 4). High sea ice buttressing occurs for low values of $f_{si}$ and vice versa. Therefore, I assume it is a typo when you state that high submarine melting would be necessary for low sea ice buttressing and vice versa?

13. p9, l8: "The values": Please specify which values you mean.

14. p11, l30: The glacier's total SMB is about 30 to 40 Gt, which is half of the modelled grounding line flux past 2015. I would therefore use a word other than "stabilizing".

15. p15, l2-9: This introductory paragraph is hard to follow. Please rephrase.

16. p17, l18: This is a one-sentence paragraph.

**References**

Bondzio, J., M. Morlighem, H. Seroussi, T. Kleiner, M. Ruckamp, J. Mouginot, T. Moon, E. Larour, and A. Humbert
2017. The mechanisms behind Jakobshavn Isbræ's acceleration and mass loss: A 3-D thermomechanical model study. *Geophys. Res. Lett.*, 44.

Enderlin, E. M., I. M. Howat, and A. Vieli
2013. High sensitivity of tidewater outlet glacier dynamics to shape. *Cryosphere*, 7(3):1007–1015.

Gudmundsson, G. H., J. Krug, G. Durand, L. Favier, and O. Gagliardini
2012. The stability of grounding lines on retrograde slopes. *Cryosphere*, 6(6):1497–1505.

Joughin, I., B. E. Smith, I. M. Howat, D. Floricioiu, R. B. Alley, M. Truffer, and M. Fahnestock
2012. Seasonal to decadal scale variations in the surface velocity of Jakobshavn Isbrae, Greenland: Observation and model-based analysis. *J. Geophys. Res.*, 117:1–20.

Morlighem, M., J. Bondzio, H. Seroussi, E. Rignot, E. Larour, A. Humbert, and S.-A. Rebuffi
2016. Modeling of Store Gletscher's calving dynamics, West Greenland, in response to ocean thermal forcing. *Geophys. Res. Lett.*, 43(6):2659–2666.

Motyka, R. J., M. Truffer, M. Fahnestock, J. Mortensen, S. Rysgaard, and I. Howat
2011. Submarine melting of the 1985 Jakobshavn Isbrae floating tongue and the triggering of the current retreat. *J. Geophys. Res.*, 116:1–17.

Muresan, I. S., S. A. Khan, A. Aschwanden, C. Khroulev, T. Van Dam, J. Bamber, M. R. van den Broeke, B. Wouters, P. K. Munneke, and K. H. Kjaer
2016. Modelled glacier dynamics over the last quarter of a century at Jakobshavn Isbræ. *The Cryosphere*, 10(2):597–611.

Pattyn, F., L. Perichon, L. Durand, G.and Favier, O. Gagliardini, R. C. A. Hindmarsh, T. Zwinger, T. Albrecht, S. Cornford, D. Docquier, J. Fuerst, D. Goldberg, H. Gudmundsson, A. Humbert, M. Hutten, P. Huybrecht, G. Jouvet, T. Kleiner, E. Larour, D. Martin, M. Morlighem, A. Payne, D. Pollard, M. Rückamp, O. Rybak, H. Seroussi, M. Thoma, and N. Wilkens
2013. Grounding-line migration in plan-view marine ice-sheet models: results of the ice2sea MISMIP3d intercomparison. *J. Glaciol.*, 59 (215):410–422.

Shapero, D. R., I. R. Joughin, K. Poinar, M. Morlighem, and F. Gillet-Chaulet
2016. Basal resistance for three of the largest Greenland outlet glaciers. *J. Geophys. Res.*, 121(1):168–180. 2015JF003643.

Truffer, M. and K. Echelmeyer

    2003. Of isbræ and ice streams. *Ann. Glaciol.*, 36:66–72. International Symposium on Fast Glacier Flow, Yakutat, Alaska, Jun 10-14, 2002.

Vieli, A. and F. Nick

    2011. Understanding and modelling rapid dynamic changes of tidewater outlet glaciers: Issues and implications. *Surv. Geophys.*, 32(4-5):437–458.

---

## Referee Report (RR2)

Review of "Non-linear retreat of Jakobshavn Isbræ since the Little Ice Age controlled by geometry" by Stieger et al., submitted to The Cryosphere

Summary: The authors use a width- and depth-integrated flowline model that includes a parameterization to account for lateral ice fluxes to test the sensitivity of Jakobshavn Isbræ's long-term retreat to variations in geometry under a variety of environmental forcing scenarios. The model results suggest that the non-linear retreat of the glacier is likely due to along-flow variations in fjord width and basal topography. The time series of grounding line and terminus retreat deviate from the observations for all the prescribed climate change scenarios, likely indicating that the simple linear climate forcings used here do not capture the complexities of the actual climate change during the observation period. However, the focus of the manuscript is on the importance of geometry in modulating the response to climate change and I think the paper clearly demonstrates that geometry exerts a strong first-order control on the timing and magnitude of dynamic change.

Specific Comments:
There are a few points that I feel should be slightly expanded on in the text for the sake of clarity and transparency in methodology.
1) It would be helpful to include an equation to clearly show how the height of basal crevasses is estimated from tensile deviatoric stresses and the height above buoyancy. There is presently no reference provided and it is up to the reader to search for an appropriate reference and equation therein that would relate these variables.
2) The transition height for the SMB parameterization is not listed in Table 1.
3) I do not see how it is possible that the ice thickness can be uniformly decreased due to submarine melting seaward of the grounding line without introducing an artificial step decrease in ice thickness across the grounding line. Is the time step sufficiently short that the step reduction in thickness at any given time is minimized? Or is submarine melting applied orthogonal to the floating ice so that it essentially ablates ice horizontally at the grounding line? Both Motyka et al., J. Geophys. Res. doi:10.1029/2009JF001632, 2011 and Enderlin et al., J. Glaciol., doi: 10.3189/2013JoG12J049, 2013 provide modern estimates of submarine melting beneath Jakobshavn's floating tongue. How do your melt rates compare? This should be stated in the text.
4) Where is the lateral influx prescribed? Is it evenly added along the lowermost 80km or is the flux weighted so that it increases or decreases in the along-flow direction? What velocity data are used for the initial parameterization? Is the average annual velocity at each grid point bordering the main trunk used to estimate lateral flux variability along flow? Please elaborate.
5) You state that a crevasse water depth of 160m during the LIA may be exaggerated but I think you should at least say it is "likely" exaggerated because it is highly unlikely that crevasse water depths are anywhere close to that deep, especially given that there is no visible water in crevasses immediately inland of the modern terminus.
6) At the bottom of page 14 you state that geometry can delay the response of glaciers to climate change. The influence of geometry on the timing and magnitude of dynamic

change was also discussed in Enderlin et al., The Cryosphere, doi:10.5194/tc-7-1007-2013, 2013 and should be cited here as support for the importance of geometry on dynamic change (albeit using simple, synthetic glacier geometries).

---

## Author Response (AR2)

Dear Olaf,

As we believe that the reviews given so far in the discussion of our paper oversee important and novel aspects of our study especially directed towards a broader community, we would much appreciate inviting a third independent reviewer of our paper.

In particular, we are referring to the lack of attention to the long term history of glaciers such as Jakobshavn and their response to climate on timescales longer than a few decades (which is clearly novel), and the fact that the reviews are very much focused on the application of heavier 3D models (which also have many caveats) rather than idealised physical models testing the underlying mechanism and their relative importance on longer timescales.
We would very much appreciate the review and comments from colleagues working on longer timescales that can place the work in the perspective of the geological records as well as the choice of model tools.

If possible, we would also ask for the interactive discussion to be extended as Andreas Vieli has approached us and suggested he would be interested in writing a comment based on his reading of the discussion paper.

We are hoping for a positive answer before we start revising the paper.

Sincerely,
Nadine Steiger and co-authors

---

## Author Response (AR3)

[revised manuscript text omitted]

$$d_s = \frac{R_{xx}}{\rho_i g} + \frac{\rho_w}{\rho_i} d_w, \text{ with } R_{xx} = 2\left(\frac{\dot{\epsilon}_{xx}}{A}\right)^{1/n}.$$
(3)

10 as the sum of tensile deviatoric stresses and enhanced by  $R_{xx}$  and additional water pressure from melt water filling up crevasses due to the additional water pressure (Eq. 3; Nye, 1957; Nick et al., 2013). The the crevasses (Nye, 1957; Nick et al., 2010). Note that the water depth in crevasses ( $d_{cw}$ )  $d_w$  is not a physical quantity, but a forcing parameter within the calving model that links calving rates to climate and is in our experiments used as a perturbation parameter.

$$d_{sc} = \frac{R_{xx}}{\rho_i g} + \frac{\rho_{fw}}{\rho_i} d_{cw}, \text{ with } R_{xx} = 2\left(\frac{\dot{\epsilon}_{xx}}{A}\right)^{1/n},$$

15 where  $\rho_{fw}$  is the densities  $\rho_{w}$  is the density of freshwater. The tensile deviatoric stress  $R_{xx}$  is the difference between tensile stresses that pulls pull a fracture open and the ice overburden pressure. It is calculated from the longitudinal strain rate  $\dot{\epsilon}_{xx}$ through via Glen's flow law (Eq. 4).

Buttressing by sea ice is implemented as from the longitudinal stretching rate  $\dot{\epsilon}_{xx}$ , which is responsible for the opening of crevasses by

20
$$\dot{\epsilon}_{xx} = \frac{\partial U}{\partial x} = f_i A \left[ \frac{\rho_i g}{4} \left( H - \frac{\rho_s}{\rho_i} \frac{D^2}{H} \right) \right]^n \tag{4}$$

in dependency of a sea ice factor  $(f_{si})$ , which can be reduced accounting for  $f_i$ , which accounts for reduced buttressing due to weakening of ice mélangeby increasing the strain rate. The strain rate (Eq. 4) is responsible for the opening and downward-penetration of crevasses at the glacier terminus, consequently increasing calving rates... The depth of basal crevasses is calculated from tensile deviatoric stresses and the height above buoyancy (Nick et al., 2010).

25
$$\underline{xx} = \frac{\partial U}{\partial x} \underbrace{d_b}_{\sim} = \underline{f_{si}A} \underbrace{\frac{\rho_i g}{4}}_{\underline{\rho_s} - \underline{\rho_i}} \left( \underline{H} \frac{R_{xx}}{\underline{\rho_i g}} - \frac{\rho_w}{\underline{\rho_i}} \frac{D^2}{H} (H - \frac{\rho_s}{\underline{\rho_i}} D) \right)_{-}^n$$
(5)

The model uses separate parameters for water-

Table 1. List of variables, physical parameters and constants used in the model. Values for the The forcing parameters with their initial (LIA)

forcing parameters values are given in the lower part. Parameter values used for the glacier retreat experiments are listed in Table 2.

| Symbol                                                | Definition                 | Value                                       | Unit                                            |  |
|-------------------------------------------------------|----------------------------|---------------------------------------------|-------------------------------------------------|--|
| Н                                                     | glacier thickness          |                                             | m                                               |  |
| t                                                     | time                       |                                             | yr                                              |  |
| W                                                     | glacier width              |                                             | m                                               |  |
| x                                                     | along-glacier coordinate   |                                             | m                                               |  |
| U                                                     | velocity                   |                                             | ${ m myr^{-1}}$                                 |  |
| B                                                     | mass balance               |                                             | ${\rm myr^{-1}}$                                |  |
| ν                                                     | viscosity                  |                                             | Payr                                            |  |
| D                                                     | depth below sea-level      |                                             | m                                               |  |
| s                                                     | surface elevation          |                                             | m                                               |  |
| $d_b$                                                 | depth of basal crevasses   |                                             | m                                               |  |
| $d_s$                                                 | depth of surface crevasses |                                             | m                                               |  |
| $R_{xx}$                                              | tensile deviatoric stress  |                                             | Ра                                              |  |
| $\dot{\epsilon}_{xx}$                                 | longitudinal strain rate   |                                             | ${ m myr^{-2}}$                                 |  |
| $Q_L$                                                 | lateral ice flux           |                                             | ${\rm myr^{-1}}$                                |  |
| a                                                     | surface mass balance (SMB) |                                             | ${\rm myr^{-1}}$                                |  |
| $s_0$                                                 | transition height for SMB  | 1600                                        | m                                               |  |
| g                                                     | gravitational acceleration | 9.8                                         | ${ m myr^{-1}}$                                 |  |
| $ ho_i$                                               | ice density                | 900                                         | ${\rm kgm^{-3}}$                                |  |
| $ ho_s$                                               | ocean water density        | 1028                                        | ${\rm kgm^{-3}}$                                |  |
| $ ho_w$                                               | fresh water density        | 1000                                        | ${\rm kgm^{-3}}$                                |  |
| m                                                     | sliding exponent           | 3                                           |                                                 |  |
| n                                                     | Glen's flow law exponent   | 3                                           |                                                 |  |
| A                                                     | rate factor taken from     | $\mathrm{A}(\text{-}20^{\circ}\mathrm{C})-$ | $\rm yr^{-1}Pa^{-3}$                            |  |
|                                                       | Cuffey and Paterson (2010) | $A(\text{-}5^{\circ}\mathrm{C})$            |                                                 |  |
| $A_s$                                                 | basal resistance parameter | 120                                         | $\mathrm{Pa}\mathrm{m}^{-2/m}\mathrm{
[revised manuscript text omitted]